# Somatosensory neurons integrate the geometry of skin deformation and mechanotransduction channels to shape touch sensing

Alessandro Sanzeni[1,2], Samata Katta[3], Bryan Petzold[4], Beth L Pruitt[4,5], Miriam B Goodman[6], Massimo Vergassola[1]*

[1]Department of Physics, University of California, San Diego, La Jolla, United States; [2]National Institute of Mental Health Intramural Program, National Institutes of Health, Bethesda, United States; [3]Neuroscience Program, Stanford University School of Medicine, Stanford, United States; [4]Department of Mechanical Engineering, Stanford University, Stanford, United States; [5]Department of Bioengineering, Stanford University, Stanford, United States; [6]Department of Molecular and Cellular Physiology, Stanford University School of Medicine, Stanford, United States

**Abstract** Touch sensation hinges on force transfer across the skin and activation of mechanosensitive ion channels along the somatosensory neurons that invade the skin. This skin-nerve sensory system demands a quantitative model that spans the application of mechanical loads to channel activation. Unlike prior models of the dynamic responses of touch receptor neurons in *Caenorhabditis elegans* (Eastwood et al., 2015), which substituted a single effective channel for the ensemble along the TRNs, this study integrates body mechanics and the spatial recruitment of the various channels. We demonstrate that this model captures mechanical properties of the worm's body and accurately reproduces neural responses to simple stimuli. It also captures responses to complex stimuli featuring non-trivial spatial patterns, like extended or multiple contacts that could not be addressed otherwise. We illustrate the importance of these effects with new experiments revealing that skin-neuron composites respond to pre-indentation with increased currents rather than adapting to persistent stimulation.
DOI: https://doi.org/10.7554/eLife.43226.001

**\*For correspondence:**
massimo@physics.ucsd.edu

**Competing interests:** The authors declare that no competing interests exist.

## Introduction

The sense of touch is a prime example of mechanotransduction in biology (*Chalfie, 2009*; *Hoffman et al., 2011*; *Katta et al., 2015*; *Schneider et al., 2016*) that culminates in the activation of mechanically-gated ion channels arrayed along sensory dendrites. These dendrites are not isolated from the tissues they innervate, nor are the relevant ion channels isolated from the extracellular matrix, plasma membrane, or the underlying cytoskeleton. In mammals, slowly-adapting mechanoreceptors depend on their association with keratinocyte-related Merkel cells for their response dynamics (*Woo et al., 2014*; *Maksimovic et al., 2014*; *Ikeda et al., 2014*). Similarly, the response dynamics of rapidly-adapting mechanoreceptors are sensitive to their association with Pacininan corpuscles (*Loewenstein and Mendelson, 1965*). Extracellular links are present in the sensory neurons innervating hair follicles (*Li and Ginty, 2014*) and in dorsal root ganglion neurons in culture (*Hu et al., 2010*). Such protein tethers are also thought to be essential for mechanotransduction by vertebrate hair cells (*Sakaguchi et al., 2009*; *Fettiplace, 2017*). The NompC channels that mediate

mechanosensation in *Drosophila* are directly linked to microtubules and, in campaniform receptors, this intracellular protein tether is essential for mechanosensitivity (*Sun et al., 2019*; *Zhang et al., 2015*; *Liang et al., 2013*). Thus, independent of their specific anatomy or mechanosensory function, the mechanically-gated ion channels that decorate sensory dendrites are intimately connected to surrounding tissues.

In *C. elegans*, the Touch Receptor Neurons (TRNs) are embedded in the skin and attached to a specialized extracellular matrix, and this structure is required for the proper distribution of MEC-4-dependent Mechano-electrical Transduction (MeT) channels (*Lumpkin et al., 2010*; *Emtage et al., 2004*). The TRNs and the MEC-4 channels enable these roundworms to evade predatory fungi that trap nematodes in a noose-like structure (*Maguire et al., 2011*). This escape behavior can also be elicited manually by drawing an eyebrow hair across the animals body (*Chalfie, 2014*) or using mechanical stimulators (*Petzold et al., 2013*; *Mazzochette et al., 2018*). These observations suggest that laboratory stimuli are sufficiently good replicas of natural stimuli that they elicit the same behaviors. Forces in the nano- to micro-Newton range are sufficient to elicit this escape behavior (*Petzold et al., 2013*; *Mazzochette et al., 2018*) in wild-type animals. Sensitivity also depends on body stiffness such that larger forces are needed to trigger escape behaviors in stiffer animals (*Petzold et al., 2013*). Thus, touch sensitivity is a combined property of the skin-nerve tissue systems.

Delivering a touch by pushing a flexible probe against the worms body activates MEC-4-containing channels (*O'Hagan et al., 2005*), connecting touch stimulation directly to activation of a specific ion channel in living animals. This process of sensory mechanotransduction depends more on the depth of body indentation than it does on the force applied (*Eastwood et al., 2015*), reinforcing the importance of tissue mechanics in touch sensation. As found for the dendrites innervating Pacininan corpuscles, the TRNs depolarize in response to the application and removal of a simple touch (*O'Hagan et al., 2005*). In (*Eastwood et al., 2015*), we introduced a simplified, but quantitative description of sensory mechanotransduction that recapitulated this on/off response dynamic. This initial model introduced a hypothetical elastic tether connected to the channel that would be stretched in response to stimulus onset, relax during continued stimulation, and stretch in the opposite direction following stimulus offset. It was inspired by the tip-link model for auditory hair cells (*Howard and Hudspeth, 1987*; *Hudspeth, 2014*), but differs from this classical model in that it posits a tangential, rather than vertical, stretching of a tether. The picture emerging from this model replaces the hinged trapdoor of hair cell mechanotransduction with a sliding trapdoor. The tangential motion emerges from the mechanics of thin shells (*Landau et al., 1986*; *Ventsel and Krauthammer, 2001*; *Audoly and Pomeau, 2011*) and applies to the worm's body and its TRNs based on their anatomical position within the animal's skin (outer shell).

While appealing in its simplicity, the model in *Eastwood et al. (2015)* is incomplete: we replaced the ensemble of channels known to be distributed along TRN dendrites by a single effective channel. This simplification is similar in spirit to a mean-field approximation in physics and shares its utility for insight as well as its theoretical and predictive limitations. An important theoretical limitation was the neglect of the nonlinear mechanics of the worm's body. Additionally, the response to variations in contact areas or stimulus timing are not well described in this simplified model. The main object of the present study is to introduce a comprehensive and quantitative description linking focal mechanical stimuli to activation of single mechanically-gated ion channels, taking into account non-linear mechanics and the spatial distribution of mechanically-gated ion channels. We evaluate the current model against prior experimental data, generating new insight into the contribution of internal hydrostatic pressure to touch sensitivity and as a major source of variation in experimental data. Additionally, we show that pre-indentation increases the response to subsequent indentation steps and use the model to reveal that this unexpected finding can be explained by the spatial distribution of tissue deformation and MeT channels. The approach underlying the present model and its evaluation by comparing simulated and experimental data could be adapted to other mechanosensory neurons that differ in their anatomy, their encapsulating tissues, and the distribution of MeT channels within the dendrites.

# Results

To improve understanding of mechanosensory transduction during touch at the systems level, we develop a comprehensive and quantitative description of how a focal mechanical stimulus or touch activates single MeT channels in *C. elegans* touch receptor neurons, taking their spatial distribution into account. The following sections include models of how touch is transformed into skin deformation and how deformation activates single MeT channels as well as comparisons between predicted and experimental mechanoreceptor currents.

## Non-linear mechanics of the nematode body and its role in converting touch into mechanical strain within the skin

A nematode's body is a tapering cylinder (*Figure 1*) that consists of an outer and an inner tube separated by a fluid-filled pseudocoelom. The outer shell includes the cuticle, skin (hypodermis), excretory system, neurons and body wall muscles, and the inner shell is formed by the pharynx, intestine and gonad (*Altun and Hall, 2009*). Adult *C. elegans* hermaphrodites are about 1 mm in length and $50 \mu m$ in diameter, at their widest point. This simple body plan has inspired models of its mechanics consisting of a cylindrical outer shell and internal pressure, which is conferred by the combined effects of internal organs and the pseudocoelom (*Park et al., 2007*). Small punctures in the cuticle and skin are thought to decrease, but not eliminate internal pressure. Indeed, this maneuver has been demonstrated to decrease stiffness inferred from force-indentation curves derived from experiments using self-sensing microcantilevers (*Park et al., 2007*; *Eastwood et al., 2015*). These and other experiments use glass probes or microbeads with a radius 5–10 µm to indent the body to a maximum depth of ~10 µm, that is about half the radius of the shell and larger than its thickness ~1 µm.

The simplest physical model consistent with the above observations is that the strain within the shell is small, so that Hookean elasticity applies, yet the displacement of material points can be of the same order or larger than the shell thickness (see Appendix 1 for details). The latter implies that the linear approximation of the strain is not appropriate and must be replaced by the nonlinear Green-Lagrange expression:

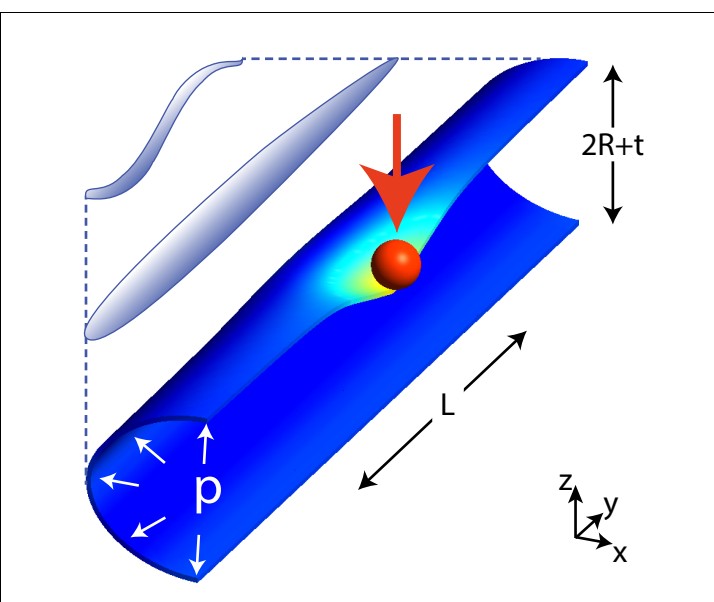

**Figure 1.** Scheme of the geometry in our model for *C. elegans* mechanics. The figure shows the scheme of a worm in a natural posture (left), straightened (as in neurophysiology experiments), and the model (right) that we shall consider here: a cylinder of length $L \simeq 1\,mm$ and radius $R + t/2 \simeq 25 \mu m$ is indented by a spherical bead (with radius 10 µm unless stated otherwise), applied here at its center. $R$ is the radius of the middle surface and $t$ is the thickness of the shell. Only half of the cylinder is shown for clarity.
DOI: https://doi.org/10.7554/eLife.43226.002

$$2\varepsilon_{ij} = \partial_i u_j + \partial_j u_i + (\partial_i u_k)(\partial_j u_k), \tag{1}$$

where $\partial_j$ is the derivative with respect to the spatial coordinate $x_j$ ($j = 1, 2, 3$) and $u_i$ is the $i$-th component of the displacement. The equations for the stress tensor $\sigma_{ij}$ (Piola-Kirchoff of the second type) read (*Landau et al., 1986*; *Ventsel and Krauthammer, 2001*; *Audoly and Pomeau, 2011*):

$$\partial_j \left[ \sigma_{ij} + \sigma_{kj} \frac{\partial u_i}{\partial x_k} \right] = 0, \tag{2}$$

where sum over repeated indices is implied. Quadratic terms in *Equations (1) and (2)* make them appropriate for large deformations. Nonlinearities are called 'geometric' due to their relation to the shape of material elements (*Ventsel and Krauthammer, 2001*; *Audoly and Pomeau, 2011*). Linear approximations that neglect geometric nonlinearities can lead to substantial discrepancies, as shown below.

The linear Hookean relation between stress and strain $\sigma_{ij} = \frac{E}{1+\nu} \left( \varepsilon_{ij} + \frac{\nu}{1-2\nu} \varepsilon_{kk} \delta_{ij} \right)$ is consistent even for large deformations provided components of the strain tensor stay moderate, which will be the case here (see Appendix 1). Here, $E$ is Young's modulus, $\nu$ is the Poisson ratio, and $\sigma_{ij}$ is energy conjugate to $\varepsilon_{ij}$, that is the elastic energy $\mathcal{E}_{el}$ upon a deformation varies as $\delta\mathcal{E}_{el} = \int \sigma_{ij} \delta\varepsilon_{ij} \, dV$.

In addition to $E$ and $\nu$, parameters of the model are (see *Figure 1*): the length $L$ and the thickness $t$ of the shell, the radius $R$ of its middle surface, and the internal pressure $p$. The pressure $p$ is understood to be the difference between the internal and the external atmospheric pressure. To simplify the following dimensional analysis, we shall neglect the external atmospheric pressure, which has only minor effects for our thin cylindrical shells (see Appendices 2 and 3). Elastic parameters are effective quantities that subsume different contributions in the inner and outer tube. Previous estimates of those parameters are discussed in the Section on Results for experimental validations. Boundary conditions generated by the pressure $p$ and the external forces are in the next Section. Finally, Appendix 4 discusses the reduction of the 3D *Equation (2)* to the 2D thin-shell limit.

The nonlinear structure of *Equation (2)* hampers analytical approaches, which pushed us to apply numerical finite-element methods (see *Burnett, 1985* for an introduction). Numerical simulations of *Equation (2)* were performed by the open-source program *code-aster* (*Électricité de France, 2001*). An hexahedral element with eight standard nodes (HEXA8) was used in combination with a mesh sensitivity analysis to verify that results are minimally sensitive to the element size. The numerical procedure was benchmarked and tested by comparing its results to known elasticity problems. In particular, Appendix 5 reports on the comparison with the deformation field and the force-indentation relation produced in small indentations of cylindrical shells, where an analytical solution is available (*Morley, 1960*), as well as large indentations of pressurized spherical shells, where a simplified equation was derived in *Vella et al. (2012a)*. In all cases, agreement was verified.

For the internal pressure $p$, active readjustments of the internal pressure are possible and could *a priori* be accommodated in our approach. Here, we shall make the simplest working hypothesis that $p$ holds constant when the stimulus is exerted onto the body of the worm. Its justification is empirical, that is we argue that the simplest option is sufficient based on results reported below.

As for boundary conditions, neurophysiology experiments have worms glued onto a plate and limited in the vertical displacement of their body's lower half (see Materials and methods). Since a mathematical formulation is not obvious (and elasticity has long-range effects), we tested different boundary conditions and present two of them for comparison (see Appendix 6). For the first, the lower half of the body is vertically rigid, that is the upper half of the cylinder in *Figure 1* is free to move, while the lower one is constrained to move only parallel to the plane onto which the worm is glued. For the second boundary condition, the lower half is fixed, that is allowed to move neither vertically nor laterally. Results presented in the main text are obtained with the latter condition. As for the two ends of the cylinder, we present results with vanishing lateral forces; Appendix 7 discusses the effects of plugs at the ends.

## Single MeT channel gating and its relationship to total mechanoreceptor currents

The TRN dendrites are decorated by MeT channels along their entire length (*Chelur et al., 2002*; *Emtage et al., 2004*; *Cueva et al., 2007*). We propose that these channels are activated by the

deformations described above. Because the touch-induced deformation and the channels are spatially distributed, we developed an approach that accounts for the activation of a single MeT channel by local deformation and subsequently computes their summation based on the spatial distribution of both the deformation and the channels. This model departs from previous work (*Eastwood et al., 2015*) by considering the contribution of individual channels to the total current and taking into explicit consideration the spatial features of the activation process.

The mechanism in *Eastwood et al. (2015)* posits that the dynamics of individual channels is the combination of an elastic and a relaxation (frictional) component. While various implementations may be contemplated, we shall refer for concreteness to a situation where each ion channel is connected to an elastic filament. We denote by $\mathbf{r}^{c,f}$ the undeformed positions of the channel and the tip of its elastic filament; the corresponding displacements induced by the deformation of the embedding tissue are $\Delta\mathbf{r}^{c,f}$.

The elastic component reflects the Hookean response of the filament to its stretching. Elastic energy is $V(\mathbf{x}) = \frac{k}{2}x^2$, where $\mathbf{x} = \Delta\mathbf{r}^{\mathbf{f}} - \Delta\mathbf{r}^{\mathbf{c}}$ is the elongation of the filament with respect to its undeformed configuration. The corresponding restoring force is

$$\mathbf{F}_{\text{elastic}} = -k\mathbf{x}\,. \tag{3}$$

As for the frictional component, the TRN and its channels are embedded in the medium and expected to move with it, that is $\Delta\mathbf{r}^c = \mathbf{u}(\mathbf{r}^c)$ where $u$ is the displacement in *Equation (2)*. Conversely, as the filament slides with respect to the medium, the friction force is

$$\mathbf{F}_{\text{friction}} = -\gamma\frac{d\big(\Delta\mathbf{r}^f - \mathbf{u}(\mathbf{r}')\big)}{dt}\,, \tag{4}$$

where $\gamma$ is the friction coefficient and $\mathbf{r}'$ is the undeformed position of the material point that coincides with the location of the tip, that is $\mathbf{r}^f + \Delta\mathbf{r}^f = \mathbf{r}' + \mathbf{u}(\mathbf{r}')$. Expanding $\mathbf{u}(\mathbf{r}')$ and using that gradients of $\mathbf{u}$ are small, we obtain $\mathbf{r}' - \mathbf{r}^f \simeq \Delta\mathbf{r}^f - u(\mathbf{r}^f)$, which is then inserted into *Equation (4)* to show that $\mathbf{u}(\mathbf{r}')$ can be replaced by $\mathbf{u}(\mathbf{r}^f)$.

Effects of inertia are negligible and the overdamped approximation holds at microscopic scales (*Phillips et al., 2012*), that is the sum of the forces $\mathbf{F}_{\text{friction}} + \mathbf{F}_{\text{elastic}} = 0$, which yields

$$\frac{d\mathbf{x}}{dt} + \frac{1}{\tau}\mathbf{x} = \frac{d\Gamma}{dt} \equiv \frac{d\big(\mathbf{u}(\mathbf{r}^f) - \mathbf{u}(\mathbf{r}^c)\big)}{dt}\,, \tag{5}$$

where $\tau = \gamma/k$ is the relaxation time, $\mathbf{x}$ is the extension of the filament, and $\Delta\mathbf{r}^c = \mathbf{u}(\mathbf{r}^c)$ was used. *Equation (5)* drives $\mathbf{x}$ to zero for $\Gamma$ constant, which is the basis for adaptation. *Equation (5)* is supplemented by the constraints exerted by the neural membrane around the channel, which limit the motion in the vertical direction. The constraint can be written as $\mathbf{x} \cdot \hat{\mathbf{w}}_3 \geq 0$, where, for every channel, $\hat{\mathbf{w}}_{1,2}$ span the plane locally tangential to the neural membrane while $\hat{\mathbf{w}}_3$ indicates the orthogonal direction. Specifically (see also Appendix 8), we define an orthonormal basis $\hat{\mathbf{e}}_i'$ as follows: $\hat{\mathbf{e}}_y'$ is aligned with the local direction of the (deformed) axis of the cylinder running head-to-tail; $\hat{\mathbf{e}}_z'$ is orthogonal to the neural membrane at the top of the TRN, and oriented outward; $\hat{\mathbf{e}}_x'$ is tangential to the neural membrane, along the remaining direction of a right-handed system. The bases $\hat{\mathbf{w}}_i$ are constructed by rotating the $\hat{\mathbf{e}}_i'$ appropriately. For a channel placed at the top of the TRN, the local basis $\hat{\mathbf{w}}_i$ coincides with $\hat{\mathbf{e}}_i'$. If the channel is rotated by $\theta$ along the surface of the TRN, then $\hat{\mathbf{w}}_1 = \cos(\theta)\hat{\mathbf{e}}_x' - \sin(\theta)\hat{\mathbf{e}}_z'$, $\hat{\mathbf{w}}_2 = \hat{\mathbf{e}}_y'$, and $\hat{\mathbf{w}}_3 = \sin(\theta)\hat{\mathbf{e}}_x' + \cos(\theta)\hat{\mathbf{e}}_z'$.

The dynamics of $\Gamma$ in *Equation (5)*, is obtained by the displacements $u$ calculated using the mechanical model. The relation between stretching and $\varepsilon$ is (*Landau et al., 1986*; *Audoly and Pomeau, 2011*): $\Gamma^2(t) - \Gamma^2(0) = 2\varepsilon_{ij}(t)\Gamma_i(0)\Gamma_j(0)$, where the $\Gamma$'s are again assumed to be small.

## The opening/closing dynamics of channels

Channels can be in multiple states: open, closed and several sub-conducting (*Brown et al., 2008*). Experiments and effects discussed here are captured by including a single sub-conducting state between the open and closed states ($C \rightleftharpoons S \rightleftharpoons O$). The respective probabilities $P_{c,s,o}$ obey the master equation

$$\begin{cases} \frac{dP_c}{dt} = -R_{cs}P_c + R_{sc}P_s \,, \\ \frac{dP_s}{dt} = -(R_{sc}+R_{so})P_s + R_{os}P_o + R_{cs}P_c \,, \\ \frac{dP_o}{dt} = -R_{os}P_o + R_{so}P_s \,, \end{cases} \tag{6}$$

where $R_{ij}$ are the respective transition rates, and $R_{c,o}=R_{o,c}=0$ again to minimize free parameters. The channels are posited to work at equilibrium, so that

$$R_{cs}/R_{sc} = e^{-\beta\Delta G_{sc}} \,; \; R_{so}/R_{os} = e^{-\beta\Delta G_{os}} \,, \tag{7}$$

where $\Delta G_{ij}$ is the free energy difference between the states $i$ and $j$, $\beta = 1/k_B T$, $T$ is the temperature, and $k_B$ is the Boltzmann constant (see, e.g., *Phillips et al., 2012*).

Channels are coupled to mechanics via their elastic filaments described by *Equation (5)*. Namely, the extension of the filament modulates the free energy differences among the above states of the channels:

$$\beta\Delta G_{oc} = g_0 - g_1\mathcal{F} \,, \tag{8}$$

where $g_0$, $g_1$ are dimensional constants, and $\mathcal{F}$ is the amplitude of the tangential component of $\mathbf{F}_{\text{elastic}}$ in *Equation (3)*:

$$\mathcal{F}_1 = \mathbf{F}_{\text{elastic}} \cdot \hat{\mathbf{w}}_1 \,; \quad \mathcal{F}_2 = \mathbf{F}_{\text{elastic}} \cdot \hat{\mathbf{w}}_2 \,, \tag{9}$$

where, for every channel, $\hat{\mathbf{w}}_{1,2}$ span the plane locally tangential to the neural membrane while $\hat{\mathbf{w}}_3$ indicates the orthogonal direction, as defined above.

Choices for the free energy other than *Equation (8)*, for example a quadratic dependence on $x$, are discussed in Appendix 9. The free energy of the intermediate subconductance state has a priori its own parameters. However, to reduce free parameters, its free energy is assumed intermediate between the closed and the open state in *Equation (8)*

$$\Delta G_{os} = a\Delta G_{oc} \,; \quad \Delta G_{sc} = (1-a)\Delta G_{oc} \,, \tag{10}$$

with the only additional parameter $0 \leq a \leq 1$. The ability of the model to quantitatively describe experimental data supports *Equation (10)* as a good empirical description of the free energy of the intermediate subconductance state.

Ion channels are believed to be distributed in spots ('puncta') along the neural membrane (*O'Hagan et al., 2005*). Their distribution is consistent with uniformity in the angular and longitudinal directions, while spacings between successive puncta are distributed log-normally (*Cueva et al., 2007*). For simplicity, each punctum is assumed to contain a single channel.

The current along the TRN is the sum $I = \sum_k i_k$ of the currents of individual channels. Its mean is given by

$$\langle I \rangle = i_o \sum_k P_o(k) + i_s \sum_k P_s(k) \,, \tag{11}$$

and its variance is calculated similarly (see Appendix 10). Here, $i_o$ and $i_s$ are the channel current in its open/subconducting state. $P(k)$ obey *Equation (6)*, and their rates depends on the position along the TRN via *Equation (3), (5), (8)*. The single-channel current $i_0 = -1.6 \pm 0.2\text{pA}$, as measured in *O'Hagan et al. (2005)* and *Brown et al. (2008)*; other parameters will be inferred from experimental data.

## The non-linear elastic model estimates mechanical parameters that agree with experiments

Next, we sought to determine the aspects of measured mechanoreceptor currents dynamics that are captured by this quantitative model incorporating body mechanics, single MeT channel gating, and the spatial distribution of MeT channels in touch receptor neurons. To achieve this goal, we compare simulations and experiments for both mechanics and neural responses.

The first step for a proper comparison with experiments is an appropriate choice of the no-stress state: the corresponding length, thickness and radius should be such that the pressurization of the

cylinder leads to the values relevant for experiments. In particular, if we want to keep the final (at pressure $p$) values fixed, the no-stress initial values should change as $p$ varies. This point, as well as our below results, differ from *Elmi et al. (2017)*, where an elastic model is discussed, yet the role of the no-stress state and pressure are not considered. Initial (no-stress) values are conveniently obtained by using perturbative analytical formulæ in Appendix 2, which give the variation of various quantities with $p$.

The schematic of an indented shell in our numerical simulations is shown in *Figure 1*. Note that the size of the indenter is not negligible with respect to other dimensions, and the region of contact with the cylinder is expected to change with the indentation depth (*Johnson, 1985*).

The thickness of the shell $t$ can be rescaled out, as discussed below in the analysis of bending and stretching contributions, and all geometric parameters appearing in *Equation (2)* are fixed. The variable factors are $p$ and $E$, which can enter the deformation for a given indentation only via their non-dimensional ratio $p/E$. We plot then in *Figure 2B* the dependence of the deformation profile along the longitudinal coordinate $vs p/E$. Vertical deformation is strongest at the center of the indenting bead, and its longitudinal extension decreases with $p/E$. The best least squares fit yields $p/E = 0.01$.

*Figure 2C* shows how the estimate $p \simeq 40 \mathrm{kPa}$ for the internal pressure was obtained: we fix the ratio $p/E = 0.01$, predict the relation force-indentation as $p$ varies, and make a best fit to the experimental data. The value of 40 kPa is on the same order of magnitude as the range of pressures measured in the larger nematode *Ascaris lumbricoides* (*Harris and Crofton, 1957*). The two above estimates yield for the Young's modulus $E \sim 4 \mathrm{MPa}$, which is the same range as the 1.3MPa obtained by measuring the bending stiffness of the nematode (*Backholm et al., 2013*) or values ~10MPa obtained in *Fang-Yen et al. (2010)* and *Petzold et al. (2011)*. Our estimate differs from the much higher values in *Park et al. (2007)*, which were also obtained by indentation data, yet using formulæ of linear elasticity that are only valid for indentations $\ll t$.

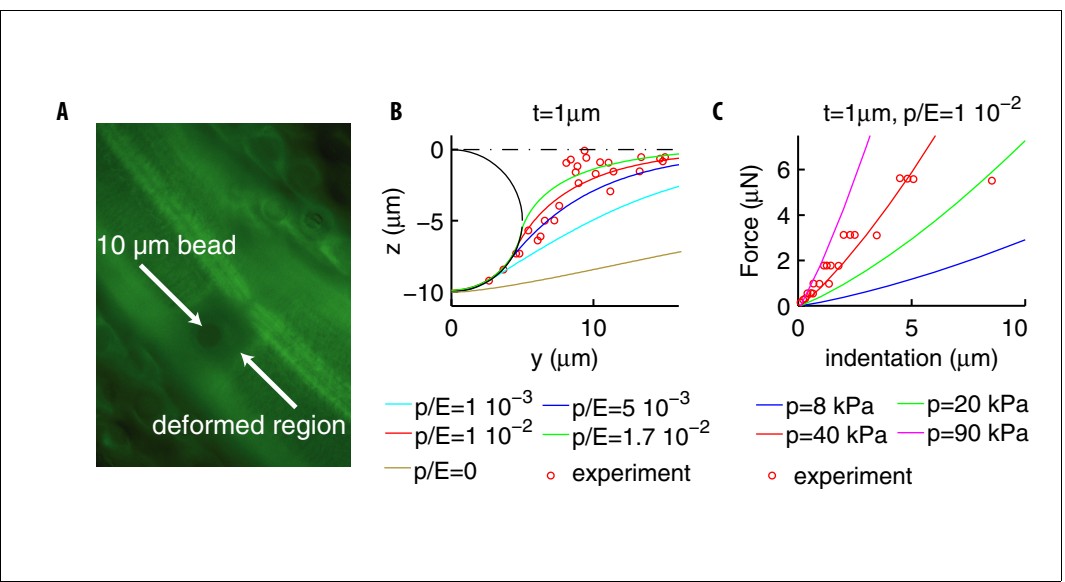

**Figure 2.** Deformation profiles and force indentation relations. (A) Representative photomicrograph of a transgenic animal with GFP-tagged cuticular annuli being pressed into a glass bead. Experimental deformation profiles in (B) were derived from a stack of images at different focal planes. (B) Experimental and numerical deformation profiles along the longitudinal axis (the generatrix of the cylinder). Data were obtained as described in Materials and methods by using 2 biological replicates (adult animals). (C) Experimental (from *Eastwood et al., 2015*; *Eastwood et al., 2019*) and numerical force-indentation relationships. Length is $L = 1mm$ and the Poisson coefficient $\nu = 0.3$.

DOI: https://doi.org/10.7554/eLife.43226.003

The following source data is available for figure 2:

**Source data 1.** Measurements of cuticle deformation by beads.

DOI: https://doi.org/10.7554/eLife.43226.004

Having fixed the parameters of our model, we can now independently test it against data on the mechanical response of *C. elegans* to changes in the external pressure (*Gilpin et al., 2015*). The variation $\Delta V$ of the initial volume $V_0$ was found to depend linearly on the variation of the external pressure $\Delta p$, and the resulting bulk modulus $\kappa = \frac{\Delta p}{\Delta V} V_0 = 140 \pm 20 \text{kPa}$. Performing the same operations in our simulations, we obtained estimated values of $\kappa = 150 - 230 \text{kPa}$. This agreement is quite significant as we derived $\kappa$, a global mechanical property, by using parameters inferred from local indentation measurements. Finally, Appendix 11 shows that mutations in the cuticle induced by disruptions of the *lon-2* gene should modify the bulk modulus, contrary to suggestions in *Gilpin et al. (2015)*.

## Predictions and experiments for responses to pre-indented stimuli

With our model reflecting the mechanical properties of the worm, we can now estimate the forces transferred to each individual channel along the TRN. By summing these individual responses to calculate the total TRN response, we can test the model's ability to explain neural responses recorded in TRNs in vivo. Predictions developed using this model recapitulate experimental responses that could not be addressed by our prior model (*Eastwood et al., 2015*), highlighting the importance of the new elements introduced here.

A detailed discussion of micro-cantilever systems of stimulation, and the in vivo patch clamp system to record neural responses was presented in *Eastwood et al. (2015)*. A summary, together with specific differences for the data first reported here, is in Materials and methods. Instances of stimuli and neural responses from *Eastwood et al. (2015)* are shown in *Figure 3* and neural responses to new, pre-indented profiles are shown in *Figure 4*.

Let us then describe how model predictions are obtained. The pressure $p$ and the parameters of the mechanical part are as in the previous Section. Positions of the channels along the TRNs were randomly generated according to the log-normal experimental distribution (*Cueva et al., 2007*), and results were averaged over those statistical realizations. As for the elastic filaments described by *Equation (5)*, their length is rescaled to unity as discussed in Appendix 12, while their initial direction $\hat{\Gamma}(0)$ is distributed randomly. Namely, directions were generated uniformly in the semisphere with a non-negative component along the local outward normal $\hat{w}_3$ to the neural membrane. Based on the deformation field determined by *Equation (2)*, we used *Equation (5)* to compute the force on the channels as a function of time, and obtained the dynamics of the channels via *Equation (6)*. More details on the fits and the resulting values of the parameters are in Appendix 12.

Our results in *Figure 3* manifestly capture the symmetric and rapidly adapting response of TRNs. Because of the onset-offset symmetry of the touch response, the response to sinusoidal stimuli

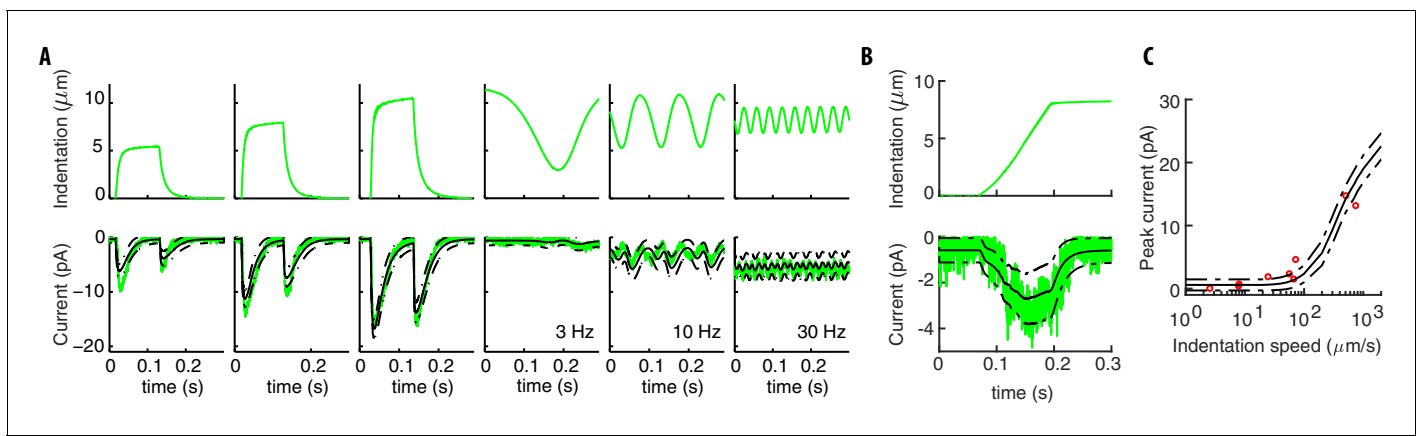

**Figure 3.** Our model captures experimental neural responses to various stimuli. (A) The applied experimental indentation (top); TRN's response (bottom, green) and average predictions (solid black). Dot-dashed black lines correspond to one standard deviation above/below the mean. Experimental stimuli and neural responses are from *Eastwood et al. (2015)* and *Eastwood et al. (2019)*. (B) A typical ramp-like profile of indentation (top) and the corresponding current (TRN's response in green; black lines as in panel A). (C) The predicted peak current vs the slope of the ramp for a total fixed indentation of 8 μm. Red circles indicate experimental data from *Eastwood et al. (2015)* and *Eastwood et al. (2019)*.
DOI: https://doi.org/10.7554/eLife.43226.005

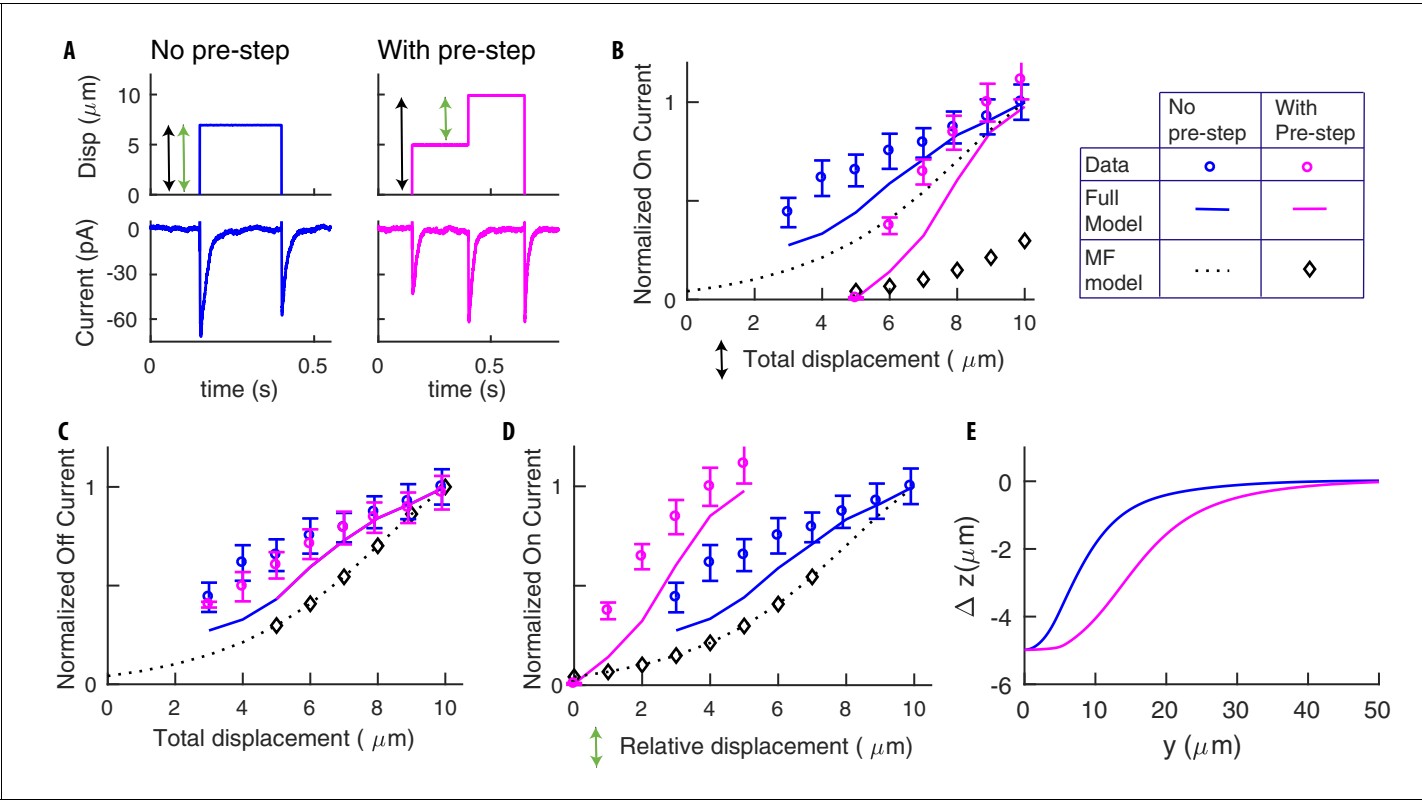

**Figure 4.** Pre-indented steps yield stronger responses due to their more extended deformation profile. (A) Stimuli delivered for standard (blue) and pre-indented (purple) steps. Black arrows indicate the total displacements for the on-currents in the following panels (colors match). Green arrows indicate relative displacements. Experimental stimuli and neural responses from ALM neurons were obtained as detailed in Materials and methods. We recorded from 11 separate worms with 3 − 11 presentations of each stimulus per recording. Recordings were only included if they met the criteria outlined in the Data Analysis section of the experimental methods, which led to a final number of biological replicates per displacement point that varied from 5 to 11. Representative traces shown here are from one biological replicate. (B) The on-current vs the total displacement (the pre-indentation for the purple points is 5 μm). Dotted curves and diamonds (in this panel and the following ones) report the prediction of our previous Mean Field (MF) model in *Eastwood et al. (2015)*. The goal is to stress the importance of spatial integration effects, which constitute the main contribution of this paper and were neglected in *Eastwood et al. (2015)*. (C) Off-currents are statistically indistinguishable, as expected since off-steps are identical and adaptation erased the memory of the pre-step. (D) The on-current vs the relative displacements. Note the stronger response for pre-indented stimuli. (E) Changes in the profile of deformation: $\Delta z$ is the difference between the deformations after and before the (relative) stimuli. Note the greater extension for the pre-indented case, which is the reason underlying results in panel D. Open circles in panels B-D were normalized to the maximal currents detected and show the mean ± s.e.m.

DOI: https://doi.org/10.7554/eLife.43226.006

The following source data is available for figure 4:

**Source data 1.** Experimental and simulated neural responses to pre-indented steps.
DOI: https://doi.org/10.7554/eLife.43226.007

oscillates at twice the input frequency. At high frequencies, inertia in the switch between open and closed states of the channels contributes to the reduced amplitude of the oscillations. The response to ramps is intuitive: the slower the indentation, the smaller the response because of adaptation. Simulations also capture the empirical relationship between speed and current amplitude (*Figure 3c*).

Appendix 13 presents the histogram of the errors for individual realizations, which shows that neural responses are captured at that level as well (not just the mean, as in *Figure 3*). The histogram also shows that restricting filaments to be initially or permanently tangential ($\Gamma(0) \cdot \hat{\mathbf{w}}_3 = 0$ or $\mathbf{x}(t) \cdot \hat{\mathbf{w}}_3 = 0$), further improves results. The latter restrictions being speculative at this stage, we shall focus on the unrestricted model; we only note that tangential restrictions admit plausible molecular mechanisms, for example. by microtubules that run along TRNs, are attached to the neural membrane through filaments and are known to impact touch sensation (*Bounoutas et al., 2009*).

Additional insight is gained by delivering stimuli alternative to the classical profiles in *Figure 3*, namely pre-indented stimuli in *Figure 4*. Panel A contrasts standard and pre-indented steps, that is where an initial step (5 µm in our data) is delivered. The neural response to two steps of equal amplitude, one pre-indented and the other not, is substantially stronger for the former. That is surprising at first, since the amplitude of the steps is identical, and enough time between the successive half steps was left for adaptation. The explanation was obtained by using our model: it is indeed the case that channels adapt and return to their rest state; however, the tissue is deformed by pre-indentation, which leads to a more extended region of stimulation and more channels activated, as shown in *Figure 4E*. The previous mean-field model in *Eastwood et al. (2015)* was unable to account for this increase in the number of channels reached by stimulation with pre-indentation (*Figure 4B,D*). The resulting predictions reproduce experimental trends, highlighting the importance of the coupling between mechanics and channel activation that constitutes the main focus of our paper.

## Variance in residual internal pressure following dissection accounts for variation in responses among individual worms

The full model not only allows us to predict neural responses to complex stimuli, but also to delve further into how body mechanics can explain the variation in experimental responses. The dissection procedure required for recording from TRNs in vivo necessarily alters body mechanics: a small incision allows some portion of the tubular internal organs (intestine and gonads) to be released outside the animal. The cuticle is largely re-sealed by the remaining internal pressure pushing large organs over the hole, and a second incision is then made to release the TRN cell body without other organs. This standard procedure results in 'soft' worms with varying fractions of their internal organs released. A modified dissection also used in *Eastwood et al. (2015)* omits the first incision and release of organs, resulting in 'stiff' worms. Names stem from the force-indentation curves in *Figure 5A*, which evidences that the latter procedure better preserves the body's integrity. Most experimental recordings reported here are for 'soft' worms while data for 'stiff' worms in *Figure 5A* were used to predict the ratio $p/E$ (*Figure 2C*). *Eastwood et al. (2015)* empirically showed that neural responses for soft and stiff worms are similar for displacement-clamped stimulations, while they strongly differ for force-clamped protocols. This Section analyzes the mechanical consequences of the above procedures, their effects on neural responses, and explains the empirical observation in *Eastwood et al. (2015)*.

Since internal organs of soft worms are removed away from the stimulation point, it is plausible that the dissection affects the internal pressure and has weaker effects on the indented external shell. This suggests to conservatively keep (in our model) the Young's modulus $E$ and the thickness $t$ fixed, and modify $p$. Results of the corresponding simulations are shown in *Figure 5A*. The slope of the force-indentation relation decreases with $p$; the best fit for soft worms is $p$ = 1.6 kPa, which is ~4% of the value for stiff worms. Finally, the scatter around the mean shows that the more invasive dissection procedure results in stronger variability, with the corresponding $p$ ranging from 0.04 to 16 kPa.

To further analyze the effects of the dissection procedure, *Figure 5B* shows the longitudinal profiles of vertical deformation for soft and stiff worms. The point is that the curves differ much less when displacement is clamped, rather than force. That is translated into predictions for currents as follows. We assume that the dissection procedure does not affect the channels and calculate their respective currents as described previously. Specifically, we fix the distribution of the channels, change $p$ to the value corresponding to soft or stiff worms, and compute the neural response to force or displacement-clamped stimuli. Results are shown in *Figure 5C*: responses for force-clamped stimuli widely differ for soft and stiff animals, yet they are similar for displacement-clamped stimuli. In sum, empirical observations reported in *Eastwood et al. (2015)* are explained by the mechanics of the nematode and its coupling to neural activation.

Finally, we address the variability in *Figure 5C* among soft worms, which we tentatively related to $p$ varying over three orders of magnitude. For further support, we tested whether the observed variability could indeed be reproduced by keeping all parameters fixed but $p$. Results in *Figure 5D* show that the peak current increases systematically with $p$ for displacement-clamp and has the opposite behavior for force-clamped stimuli. The predicted change is larger in the former case, which is consistent with differences between soft and stiff worms. The model predictions are compared with

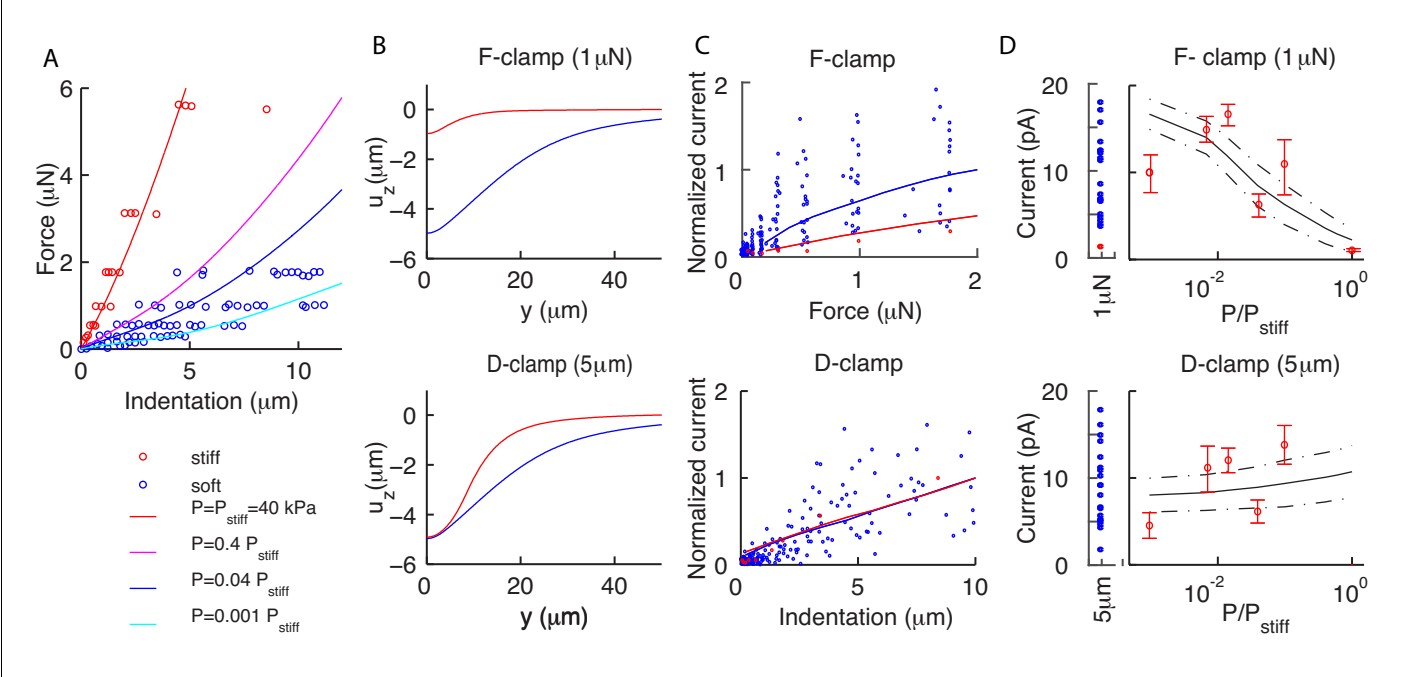

**Figure 5.** Residual internal pressure accounts for current amplitude in soft and stiff worms. (**A**) Experimental data (dots) and the average theoretical prediction (lines) for force-indentation relations. Best fit of pooled data for soft worms gives $p = 1.6$ kPa; individual values are variable, with estimated $p$ in the range 0.04–16kPa. (**B**) The vertical deformation profiles $u_z$ vs the position $y$ along the longitudinal axis for stiff (red) and soft (blue) animals. Note the widely differing profiles for the force-clamped curves. (**C**) Experimental (dots) and theoretical (mean value as continuous lines) peak current for force (top) and displacement-clamped (bottom) stimuli. The current is normalized by the mean peak in soft and stiff worms, respectively. (**D**) Peak current vs the pressure $p$, which shows that the model (continuous lines are the mean; dot-dashed lines are above/below one standard deviation) captures experimental trends (dots). Experimental data reproduced from *Eastwood et al. (2015)* and *Eastwood et al. (2019)* and derived from 4 and 21 recordings in the stiff (red) and soft (blue) conditions, respectively.

DOI: https://doi.org/10.7554/eLife.43226.008

The following source data is available for figure 5:

**Source data 1.** Experimental and simulated neural responses to force-clamped or displacement-clamped stimuli.

DOI: https://doi.org/10.7554/eLife.43226.009

an experimental dataset of 21 worms obtained in *Eastwood et al. (2015)*. For each worm we inferred $p$ from the force-indentation relation, pooling together animals with similar trends. *Figure 5D* supports the initial hypothesis that differences in $p$ among dissected animals are a major component in the observed variability.

## Testable model predictions for future experiments

In the previous sections, we were able to validate many of our predictions using existing or new data, increasing our confidence in the model. The following subsections illustrate how the model can be used to make predictions to be tested in future experiments.

### Shell bending is weak compared to stretching; stiffness is dominated by internal pressure

The mechanics of pressurized shells relies on the balance between the internal pressure $p$, bending and stretching of the shell. Contradictory results have left undecided the previous balance for *C. elegans* (*Park et al., 2007*; *Gilpin et al., 2015*). Here, we exploit our model to clarify this issue.

We computed the vertical deformation for different values of $p/E$ and $t$, as previously done for the validation of the mechanical model. The longitudinal extension is quantified by the distance $y_h$ for the deformation to reduce to half of its maximum value (at the center of the bead). *Figure 6*

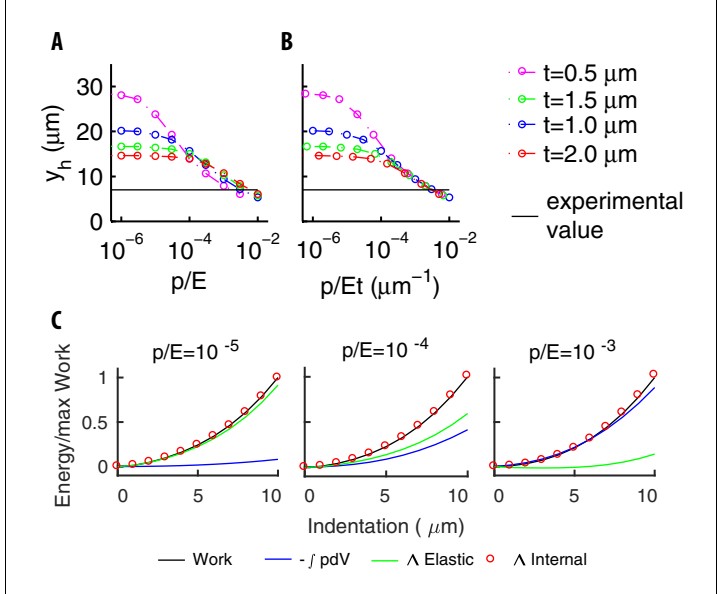

**Figure 6.** The mechanical balance for our model of pressurized shell. (**A**) The longitudinal extension $y_h$ vs $p/E$ for various thicknesses $t$. Different trends at small and large values of $p/E$ reflect the contributions of bending to the elastic energy. (**B**) $y_h$ vs $p/Et$. The collapse of the curves at the right end reflects the small value of the bending term coefficient (see the text). The value of $y_h$ found experimentally (black line) is well inside that asymptotic region. (**C**) The various contributions to energy, and the work done by the indenter for increasing values of the ratio $p/E$.

DOI: https://doi.org/10.7554/eLife.43226.010

shows that $y_h$ decreases, that is the deformation is more localized, when $p/E$ increases. Conversely, as $t$ reduces, the deformation is wider if $p/E \lesssim 10^{-4}$ and narrower if $p/E \gtrsim 10^{-4}$.

To gain insight regarding the consequences of *Figure 6*, we can use the thin-shell limit of *Equation (2)* in Appendix 4. Reducing the limit equation to non-dimensional form as previously done for spheres (*Vella et al., 2012a*), we find that the bending term is multiplied by the factor $1/\tau^2 \equiv E^2 t^4 / p^2 R^4$. If $\tau \gg 1$, the bending term is small, and (with the possible exception of boundary layer regions) the only remaining dependence on $t$ is via the stiffness $S \equiv Et$. These arguments suggest to plot $y_h$ vs $p/Et$ as in *Figure 6*: curves with different $t$ indeed collapse for the values of $p/Et$ that are relevant for experiments. We conclude that internal pressure and stretching of the shell provide the dominant balance.

We next compared the elastic energy of the shell with the work by the external forces. Results in *Figure 6C* show that their ratio reduces as $p/E$ increases, and the elastic energy tends to become marginal, which illustrates the dominance of the internal pressure in the body stiffness.

## Mechanical and neural responses depend on the radius of the indenting bead

Previous research has noted that the amplitude of neural currents depends on the radius $R_b$ of the indenting bead (*O'Hagan et al., 2005*), but no systematic study has been made of this relationship. We fill this knowledge gap with simulations of how bead size and internal pressure interact to affect the deformation of the worm and thus the neural currents produced.

Results are shown in *Figure 7*: $R_b$ influences both the deformation profile (*Figure 7A*) and the force-indentation curve (*Figure 7B*) for $p/E$ in the experimentally relevant range. The curves are intuited as follows. The curvature of the deformation field at the indentation point increases with $p/E$ until it matches the radius $R_b$. As $p/E$ increases further, the shell cannot become any steeper (the bead is rigid), so that it adapts to the bead in the contact region (see *Figure 2B*), which widens with $R_b$. The radius $R_b$ also controls the deformation outside of the contact region, namely the mid-

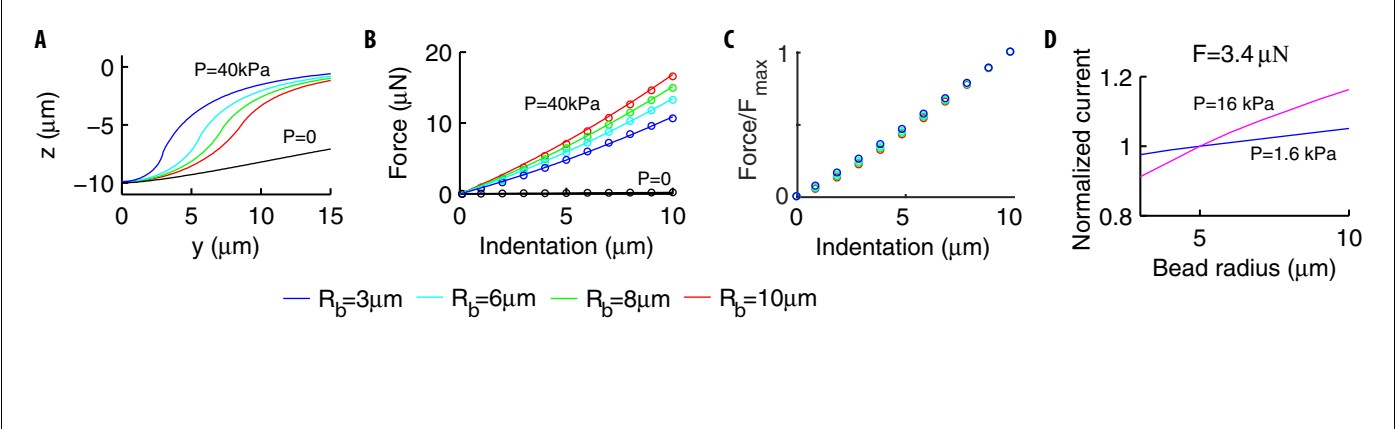

**Figure 7.** Effects of the radius $R_b$ of the indenting bead. (A) In pressurized shells, the deformation profile depends on $R_b$ (colored lines) while the dependence disappears in the absence of internal pressure (black line). (B) The force-indentation relation for $p = 40kPa$ and various $R_b$. In strongly pressurized shells, the relation (colored dots) follows **Equation (12)** (solid lines). For shells with $p = 0$, the curves collapse onto a unique curve (black dots). (C) The ratio $F(w_0)/F_{max}$ of the force normalized by its maximum value is essentially independent of $R_b$. (D) The peak current increases with $R_b$ in a $p$-dependent manner (the same holds for the sensitivity to the stimulus). The current is normalized by its value for $R_b = 5\mu m$.

DOI: https://doi.org/10.7554/eLife.43226.011

maximum extension of the deformation ($\propto R_b$, data not shown). As $p/E$ reduces, the deformation becomes shallower at the indentation point and the role of $R_b$ vanishes (see **Figure 2B**).

Similarly, **Figure 7A** shows that the volume of the body to be deformed increases with $R_b$, and more work is needed for a given maximum indentation $w_0$. In formulæ: the work $Fdw_0$ by the indenter roughly balances the contribution of the internal pressure $-pdV$ (the elastic energy is small, as discussed previously) and the force-indentation relation is then given by $F \propto -p\,dV/dw_0$.

Qualitatively, larger $dV$'s associated with larger beads yield the nonlinear dependence in **Figure 7B**. Quantitatively, we write $F = p\chi$, where $\chi = -dV/dw_0$ has the dimension of a length squared and depends on $w_0$, the shell and bead radii $R$, $R_b$, and the ratio $p/E$. Keeping the latter fixed, we investigated numerically the behavior of $\chi$ and observed that the ratio $F(w_0)/F_{max}$ does not depend on $R_b$ (see **Figure 7C**). It follows that the dependence on $R_b$ should factorize out: $\chi = \mathcal{G}_1(R_b/R)\mathcal{G}_2(w_0, R)$ where $\mathcal{G}_1$ depends on $R_b/R$ for dimensional reasons. The function $\mathcal{G}_2$ brings the length squared dimensionality and, in the limit of small $R_b$ and large $p$, behaves as $R w_0$ (**Vella et al., 2012b**). It follows that $\mathcal{G}_2 = Rw_0\mathcal{G}_3(w_0/R)$.

The above functions $\mathcal{G}_3$ and $\mathcal{G}_1$ are determined as follows. We computed numerically the force indentation relation of cylinders of different radii ($R$ = 25, 40, and 50 μm) to stimulations produced by beads of different size ($R_b$ from 3 to 10 μm); results of the simulations are then used to fit coefficients of the Taylor expansions of $\mathcal{G}_1(x)$ and $\mathcal{G}_3(x)$. Using this approach we find that the functional form

$$F = \alpha_1 pRw_0\left(1 + \alpha_2\frac{R_b}{R}\right)\left(1 + \alpha_3\frac{w_0}{R}\right), \tag{12}$$

with $\alpha_1 = 0.76$, $\alpha_2 = 2.1$, and $\alpha_3 = 0.66$, captures quantitatively the behavior of the force indentation relation (**Figure 7B**, $R^2 = 0.995$). Variations between $R_b = 3$ μm and $R_b = 10$ μm are on the order of few μN, hence they should be accessible experimentally. **Equation (12)** generalizes the linear relationship, valid in the limit of very small $R_b$, obtained in **Vella et al. (2012b)**.

Consequences for neural responses are in **Figure 7D**. In agreement with **O'Hagan et al. (2005)**, the peak current increases by ~20% as $R_b$ goes from 3 to 10 μm, hence our prediction could be tested experimentally. A quantitative comparison with data in **O'Hagan et al. (2005)** was hampered by the lack of force-indentation measurements in **O'Hagan et al. (2005)**, preventing us from inferring $p$.

Finally, it is worth remarking that the bead size also affects the dependence of the response on the circumferential position of the TRN. Indeed, **Appendix 6—figure 1** evidences that the profile of

deformation decays rapidly as one moves circumferentially from the north pole (where the bead is indenting the body) toward the equator. That implies an appreciable dependence on the angular position of the TRN with respect to the bead, which will be stronger for smaller beads as the extension of their deformation is reduced.

## Similar tangential forces at stimulus onset and offset drive symmetric on/off responses

Thus far, we have treated the fact that TRNs respond to both the application and release of a step stimulus as a given. Yet channels in many other mechanosensitive systems respond preferentially to stimuli in a particular direction - either on or off - rather than responding symmetrically to both (see *Katta et al., 2015* for review). Here, we further analyze the origin of this symmetry, by calculating the stimuli upon the channels and analyzing differences among microscopic gating mechanisms that are consistent with the symmetry.

A first key remark, which generally applies to thin shells (*Landau et al., 1986*; *Audoly and Pomeau, 2011*), is that the off-diagonal components $\varepsilon_{xz}$ and $\varepsilon_{yz}$ are small compared to the rest of the components of the tensor, namely the tangential ones. Indeed, those two off-diagonal terms are proportional to the corresponding components of $\sigma$, which vanish due to the thinness of the shell (see *Landau et al., 1986*; *Audoly and Pomeau, 2011*). It follows from the definition of $\varepsilon$ (see Appendix 8) that vectors initially tangential and perpendicular to the surface of the cylinder, remain orthogonal even after deformation.

In addition to the general above property, the component $\epsilon_{xy}$ is also negligible when the indenting bead is applied on top of the cylinder. The strain tensor is then diagonal, as confirmed by *Figure 8B*.

The force acting on a single channel, as defined by *Equation (9)* and calculated using *Equation (5)*, is shown in *Figure 8C*. The force is maximal if the elastic filament is initially in the tangential plane while orthogonal filaments generate negligible forces. Notably, forces for tangential filaments have opposite signs yet very similar amplitudes at the onset and offset. The relation between vertical and tangential directions is key to the onset-offset symmetry and stems from the above discussion on thin shells. That constitutes the physical reason for our positing that tangential stimuli gate the channels: the orthogonal dynamics is indeed affected by the neural membrane, which a priori prevents any symmetry between inward and outward extensions.

## Models with symmetric or directional channel populations could be distinguished experimentally

Though the forces reaching the channel at stimulus onset and offset are similar in amplitude, they are opposite in direction (*Figure 8B*). Two alternative mechanical models could then explain the observed symmetry: a 'symmetric' model in which each individual channel responds to force in both directions, and a 'directional' model in which individual channels respond preferentially to force in one direction, but the population as a whole responds to both. Namely, alternatively to the isotropic choice in *Equation (8)*, we could consider the 'directional' model with the preferential direction $\mathbf{v}$:

$$\beta\Delta G_{oc} = g_0 - g_2\,\mathcal{F}\cdot\mathbf{v}. \tag{13}$$

Contrary to *Equation (8)*, *Equation (13)* breaks the symmetry for individual channels (see *Figure 8D*), which can be restored though for the total current if channels along the TRN have their directions $\mathbf{v}$ in *Equation (13)* independently and isotropically distributed (*Figure 9A*).

A more quantitative analysis leans toward the symmetric model. Indeed, for the experimental value (*O'Hagan et al., 2005*; *Brown et al., 2008*) of the single-channel current $i_0 = -1.6 \pm 0.2$pA, *Equation (13)* underestimates the mean current (*Figure 9A*). More generally, we can optimize parameters and calculate the errors in the fits of the experimental datasets: the probability distribution for *Equation (13)* is broader and shifted to higher errors with respect to the isotropic model (see *Figure 9B*). Similar conclusions hold if $i_0$ is allowed to vary (*Figure 9C*).

In sum, the analysis of available experimental data favors symmetric channels, but is not fully conclusive. New data will be needed, which is our motivation to describe hereafter two possible experiments.

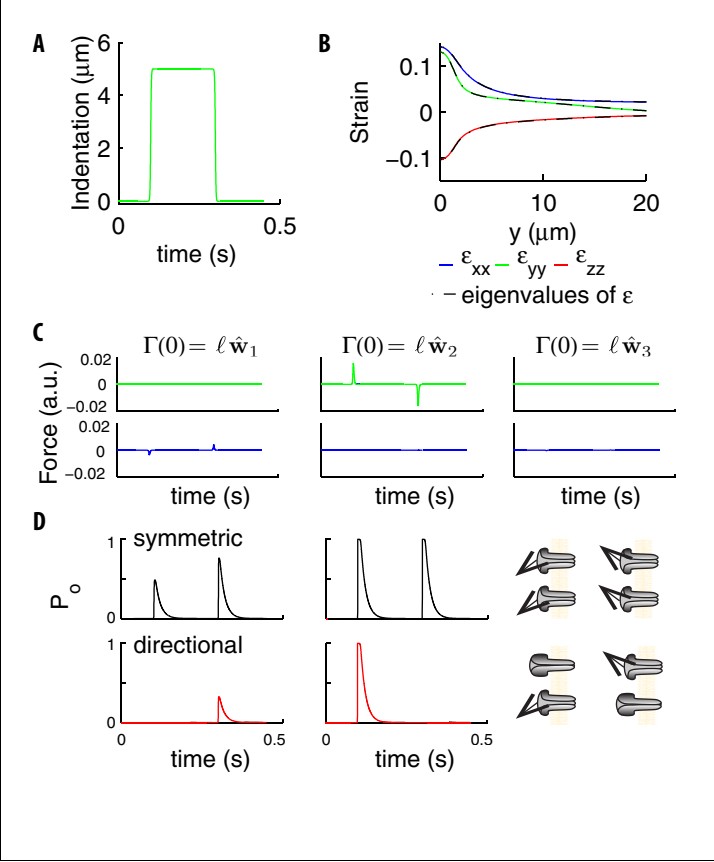

**Figure 8.** Stimuli on the channels due to a step. (**A**) Indentation profile for a step stimulus. (**B**) The diagonal components of the strain tensor $\varepsilon_{ij}$ vs the longitudinal position along the cylinder. The overlap of those components (color) and the eigenvalues of $\varepsilon_{ij}$ (black) show that the tensor is essentially diagonal, which leads to the conservation of angles under deformation discussed in the main text. (**C**) The two components (green and blue curves) tangential to the neural membrane of the force acting upon on a channel (computed using *Equations (5), (9)*) for the stimulus in panel A. The panels refer to the different directions of the elastic filament (the first two are tangential and the third orthogonal to the neural membrane). (**D**) Gating probability for an individual symmetric or directional channel, as produced by the two tangential extensions in panel C. Parameters are: $y = 1\mu m$, $\theta = 0$, $v = \cos(\pi/3)\,\hat{\mathbf{w}}_1 - \sin(\pi/3)\,\hat{\mathbf{w}}_2$. The sketch on the right illustrates that directional channels respond only to stimuli properly aligned with respect to their preferential direction while symmetric channels respond isotropically.

DOI: https://doi.org/10.7554/eLife.43226.012

A first approach relies on the noise level of currents. The intuition is that anisotropy reduces (for a given density of channels) the number of active channels along the TRN, and thereby leads to more noise. Specifically, the number of active channels could be inferred (Appendix 10 includes a generalization of the noise analysis in *O'Hagan et al., 2005* to non-equally-stimulated channels) and compared to the number of channels measured by fluorescent tags (*Cueva et al., 2007*). *Figure 9D* presents the Coefficient of Variation (CV) of the TRN current vs the stimulus strength, calculated over repetitions of a given stimulus. Differences in CVs are poised to permit discrimination and the approach described in Appendix 10 estimates that ~ 100 trials suffice for their reliable measurement.

A second alternative exploits the architecture of *C. elegans* neurons: TRNs extend longitudinally for about half of the nematode's length, leaving a region around its center that is relatively insensitive to touch (see *Mazzochette et al., 2018*). *Figure 9E* indicates the range over which effects of indentation are felt by individual channels; panel F shows the differences between microscopic models as the indenting bead slides along the longitudinal direction. An additional relevant statistic is the asymmetry between on- and off-currents. The logic is that, as the number of stimulated channels

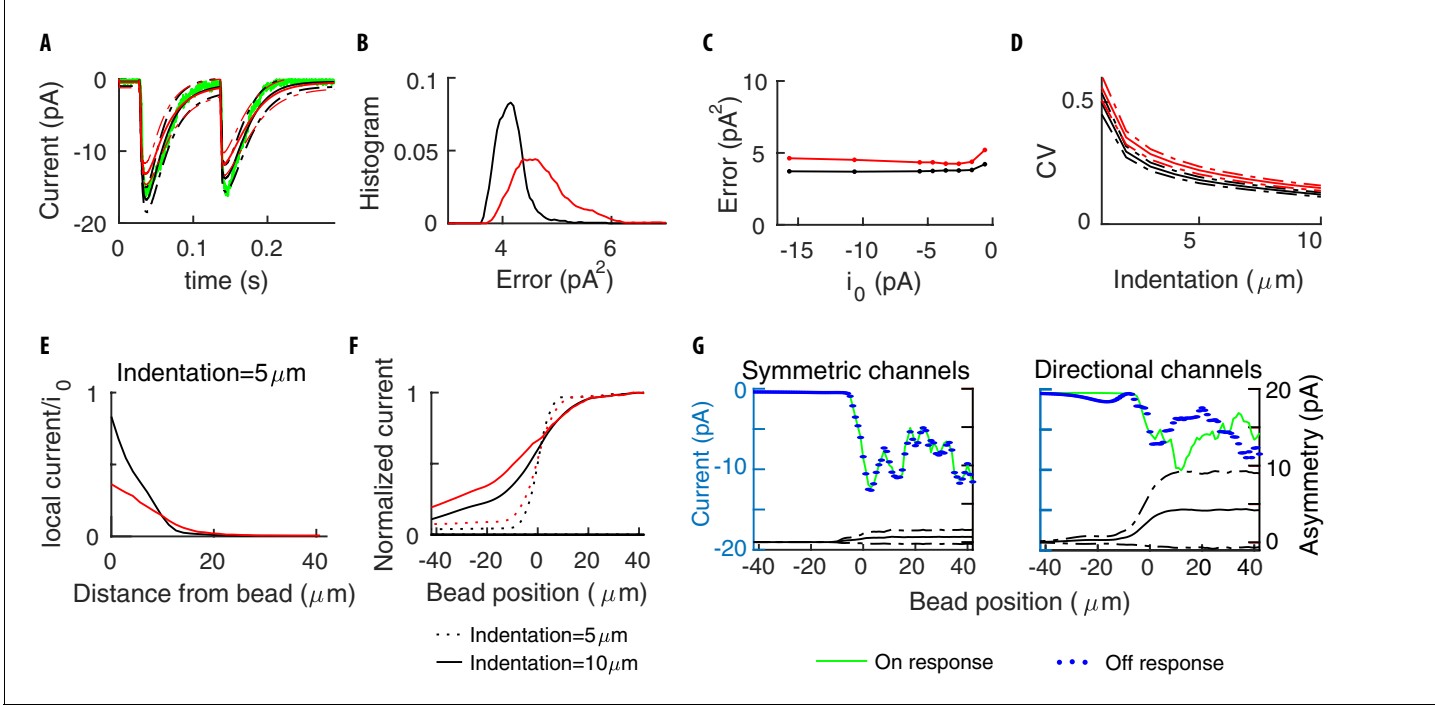

**Figure 9.** Symmetry of the single channel response. (**A**) Mean neural current response to a step, for symmetric (black) vs directional (red) channels. (**B**) Histogram of the errors in reproducing the data of *Figure 3* obtained with the two above models for different realizations of the channels' distribution. Symmetric channels (black) give a better description. (**C**) The mean error as the maximum current $i_0$ per channel is varied. (**D**) The Coefficient of Variation (CV) of the TRN current vs the stimulus strength, calculated over many repetitions of a given stimulus. (**E**) The average current for a channel (normalized by its maximum value $i_0$) as a function of its distance to the center of the indenting bead. (**F**) The current flowing along the TRN vs the position of the indenting bead. The origin indicates the end of the TRN; negative coordinates correspond to the relatively insensitive zone in the middle of the body of the worm. (**G**) The colored curves show the predicted current for symmetric and directional channels, for a given distribution of the channels. The black curves show the expected level of asymmetry between onset and offset, as quantified by the standard deviation $\left\langle \left(I_{on} - I_{off}\right)^2 \right\rangle^{1/2}$ between the peak responses $I_{on,off}$ at the onset/offset of the stimulus, averaged over the distributions of the channels. Dashed-dotted curves show the range of expected asymmetries in individual realizations.

DOI: https://doi.org/10.7554/eLife.43226.013

decreases, asymmetries should become more substantial if the channels are not isotropic, see *Figure 9G*. An appealing possibility is the stimulation of the worm in its center (negative coordinates in panels F,G). There, the number of activated channels is small, which could indeed bring microscopic insight.

## Discussion

We present a quantitative description of the response of *C. elegans* touch receptor neurons to simple and complex mechanical cues. This work combines modeling and simulations of how touch deforms the skin and its embedded TRNs with a detailed model of the activation of single MeT channels, linking skin indentation to neuronal strain and MeT activation. This model explains several facets of the coupling between tissue mechanics and neural responses that were not previously understood.

Our model explains several aspects of the coupling between mechanics and neural responses. First, the model replicates experimental observed currents evoked by a wide range of indentation profiles. The prior model (*Eastwood et al., 2015*) relied on a mean-field approximation of channel activation, which could not properly account for the fact that pre-indentation increases the number of channels that contribute to the total current. Second, the model explains how the mechanics of skin deformation contribute to the empirical observation previously made in *Eastwood et al. (2015)* that neural responses are variable for force-clamped stimulation but less so for displacement clamp.

It shows that variation in internal pressure resulting from the dissection procedure is a major source of variability in the neural responses of individual nematodes. Third, the model predicts that the neural response should increase with the size of the indenter in a manner dependent on internal pressure. Finally, the variation in deformation around the circumference of the worm suggests that the angular position of the TRN with respect to the indentation bead affects the response. These findings can help design future experiments to include measurements of these important parameters.

Our analysis also provides insight on two points related to the mechanics of the worm and the underlying biology. These findings reconcile conflicting results (*Park et al., 2007*; *Gilpin et al., 2015*) on mechanical properties in wild type and mutants. Specifically, we showed that the mechanical response of the nematode is captured by an elastic cylindrical thin shell in a pressure-dominated regime. The theory makes testable predictions on the dependence of the force-indentation relation on the indenter size, as well as the effects of mutations in the cuticle on the bulk modulus. An experimental verification of those predictions would further support the major role of internal pressure in *C. elegans* body mechanics. The fact that our model yields best results for boundary conditions with no force at the two ends of the cylinder, suggests that the body of the nematode might relax longitudinal components of the stress. One possible mechanism is through the annular structure of the cuticle (*Altun and Hall, 2009*), which may be effectively described as a shell with anisotropic Young's moduli. This relaxation of longitudinal stresses could facilitate the bending motions required for nematode motility.

We have shown how neural responses have similar amplitude when the stimulus is applied or released, which addresses the long-standing puzzle of the onset-offset symmetry. The picture emerging from our work is that the most likely gating model involves displacement tangential to the neuronal membrane. This insight can guide the search for the biophysical basis of in vivo activation of MeT channels and differs fundamentally from other models that involve orthogonal components (*Howard and Hudspeth, 1987*; *Chalfie, 2009*; *Hudspeth, 2014*). At the microscopic level, we inquired whether the symmetry holds for individual channels or at the population level only (individual channels are asymmetric yet their preferential directions are randomly distributed and their cumulative effect is again symmetric, as suggested for *Drosophila* sound receiver [*Albert et al., 2007*]). We showed, however, that a model with symmetric channels gives a better description of existing data, hinting at symmetry for single channels. Definitive evidence could be obtained by the experiments that we suggested in *Figure 9C and G*, with the noise levels better controlled and the stimulation point moved along the longitudinal axis so as to assay a variable number of channels. The ideal experiment would be to precisely assay the neural response to stimuli in the central dead-zone of the body, where few channels are likely to be directly stimulated (see *Mazzochette et al., 2018*).

In our current description, we assumed that the material composing the shell is purely elastic and the dependencies on frequency result from the gating of the channels. While this procedure successfully captures many experimental observations, it is known that tissues do feature viscous effects (*Backholm et al., 2013*). Future developments will address viscoelastic effects, which should be relevant to the understanding of touch sensation at high frequencies.

Finally, it is worth noting that our modeling ultimately relies on the fact that touch receptor neurons are close to the surface of the skin's thin layers. This leads to physical effects peculiar to thin shells, namely the importance of tangential forces, which are at the basis of the gating mechanism discussed here. Since the above features are common in touch sensation, we expect results and methods that we developed to be widely relevant.

## Materials and methods

We incorporated most of our modeling methods in Results and Discussion. Numerical simulations were performed as discussed in Results by the open-source program *code-aster* (*Électricité de France, 2001*). For additional details, please see Appendices. Experimental methods are found hereafter.

## Experimental methods

### Nematode strains

The following transgenic *C. elegans* nematodes were used: TU2769 *uls31[mec-17p::gfp]* III (*O'Hagan et al., 2005*) and TP12 *kals12[col-19::gfp]* (*Thein et al., 2003*). The corresponding identifiers are RRID:WB-STRAIN:TU2769 and RRID:WB-STRAIN:TP12, respectively. The *uls31* transgene expresses GFP exclusively in the TRNs, enabling in vivo recordings from these neurons and the *kals12* transgene encodes a fusion between the COL-19 collagen protein and GFP, labeling cuticular annuli. Animals were grown on OP50 at either 15°C (TU2769) or 20°C (TP12) and used as well-fed L4 larvae or young adults.

### Imaging Cuticle Deformation

TP12 worms were immobilized with 0.1 μm polystyrene beads on a 6% NGM agarose pad. 10 μm glass beads (Duke Scientific) for indenting the worms were spread onto a coverslip, which was inverted to cover the agarose pad holding the worms. To image the worms, we used a high-magnification camera (Orca-R2, Hamamatsu) on an inverted microscope (Leica) with an EGFP filter set and a high-numerical aperture 63x oil immersion lens, to yield a shallow depth of field ≈ 0.1 μm for optical sectioning. When glass beads were trapped between the cuticle of the animal and the coverslip, we were able to capture fluorescence images of COL-19::GFP in the cuticle at >10 different focal planes. At each focal plane, we measured the radius of the bead and the radius of the cuticle deformation (by identifying where the cuticle was in focus). We then calculated the depth of the plane based on the radius of the bead at the focal plane. Experimental data shown are a combination of all focal planes for two adult animals.

### Electrophysiology

Worms were immobilized on 2% agarose pads with WormGlu (GluStitch), dissected, and patch-clamped as described in *Eastwood et al. (2015)*. Recordings were performed on the ALMR neuron due to geometric constraints of the stimulator system; ALMR is bilaterally symmetric to the previously used ALML neuron. The extracellular solution contained (in mM): NaCl (145), KCl (5), MgCl$_2$ (5), CaCl$_2$ (1), and Na-HEPES (10), adjusted to pH 7.2 with NaOH. Before use, 20 mM D-glucose was added, bringing the osmolarity to ~325mOsm. The intracellular solution contained (in mM): K-Gluconate (125), KCl (18), NaCl (4), MgCl$_2$ (1), CaCl$_2$ (0.6), K-HEPES (10), and K$_2$EGTA (10), adjusted to pH to 7.2 with KOH. Before use, 1 mM sulforhodamine 101 (Invitrogen) was added to help visualize successful recording of the neuron.

Membrane current and voltage were amplified and acquired with an EPC-10 USB amplifier and controlled through Patchmaster software (HEKA/Harvard Biosciences). The liquid junction potential between the extracellular and intracellular solutions was −14 mV and was accounted for by the Patchmaster software. Data were sampled at 10 kHz and filtered at 2.9 kHz.

Electrophysiology source data from *Eastwood et al. (2015)* are available upon request.

### Mechanical stimulation

For mechanical stimulation during patch-clamp electrophysiology, previous studies used either open-loop systems with a piezoelectric bimorph (*O'Hagan et al., 2005*) or stack (*Bounoutas et al., 2009*; *Arnadóttir et al., 2011*; *Geffeney et al., 2011*; *Chen et al., 2015*) with no measurement of actual displacement or a closed-loop system with a stimulus bead at the end of a piezoresistive cantilever for force detection, driven by a piezoelectric stack (*Eastwood et al., 2015*). Here, we use an open-loop system adapted from the piezoelectric stack system with a photodiode motion detector described in *Peng et al. (2013)*. This enables faster stimulation than the force-clamp system (*Eastwood et al., 2015*; *Petzold et al., 2013*) at the expense of control over and measurement of exact force and indentation. The photodiode detector allows for a readout of the time course of the displacement of the stimulator.

An open-loop piezoelectric stack actuator with 20 μm travel distance (PAS-005, ThorLabs) was attached with marine epoxy (Loctite) to a 0.5'' diameter, 8'' length tungsten rod, and mounted on a micromanipulator (MP-225, Sutter) at a 17° angle to allow the stimulator to fit beneath the microscope objective.

For detecting probe motion at the 0.5–10 μm scale, we adapted the system from *Peng et al. (2013)* to use the SPOT-2D segmented photodiode (OSI Optoelectronics), and mounted it in an XY translator on top of a rotation stage (ST1XY-D, LCP02R, ThorLabs) to enable alignment of the photodiode gap perpendicular to the direction of probe motion. This was affixed above a secondary camera port on the microscope (Eclipse E600FN, Nikon) with no additional magnification.

To create a defined and reproducible contact surface for the stimulation probe, we adapted the bead gluing technique used previously for the force-clamp system (*Petzold et al., 2013*; *Eastwood et al., 2015*), but with an opaque bead that allowed for a clear signal from the photodiode motion detector. Borosilicate glass pipettes (Sutter, BF150-86-10) were pulled and polished to a tip diameter of 10–15 μm, and 20–23 μm diameter black polyethylene beads (BKPMS-1.2, Cospheric) were attached with UV-curable glue (Loctite 352, Henkel). Pipettes with attached beads were trimmed to a length of 1–2 cm, placed in the pipette holder, and waxed in place with sealing wax (Bank of England wickless, Nostalgic Impressions). A high-resolution 3D-printed acrylic pipette holder (custom design) was attached with marine epoxy to a steel tip (PAA001, ThorLabs) mounted on the piezo stack.

After cell dissection, but before making a gigaseal for patch clamp, the front edge of the stimulator bead was moved into place and visually aligned under the 60X objective with the highest visible edge of the worm's cuticle at a distance of 108 ± 36 μm anterior to the ALM cell body.

## Stimulus control and data acquisition

All systems described here were controlled through HEKA Patchmaster software with a 10 kHz sampling frequency. The voltage output from the EPC-10 amplifier (HEKA) was adjusted based on the total range of the stack for a relationship of 0.418 V/μm. This command signal was filtered at 2.5 kHz on an 8-pole Bessel filter (LPF-8, Warner Instruments) and then amplified with a high-voltage, high-current Crawford amplifier (*Peng and Ricci, 2016*) to achieve a signal between 0–75V which was sent to the stack. The stack was biased with a starting offset of 3–4 μm, and the largest displacement used was 3–4 μm less than the upper limit of the stack's travel distance, ensuring that stack motion was linear. The analog signal from the photodiode circuit was digitized at a rate of 10 kHz by the EPC-10 amplifier and Patchmaster software, for temporal alignment of the probe motion signal with the evoked current response.

## Data analysis

Whole-cell capacitance and series resistance were measured as previously described (*Goodman et al., 1998*). Data analysis was performed with MATLAB from Mathworks (data import and analysis functions are available online at: http://github.com/wormsenseLab/Matlab-Patchmaster and Igor Pro (Wavemetrics). The identifier of the MATLAB-Patchmaster analysis code is https://github.com/wormsenseLab/Matlab-PatchMaster/tree/vSanzeni2 (copy archived at https://github.com/elifesciences-publications/Matlab-PatchMaster).

Only recordings with holding current <-10pA at -60mV and series resistance <210MΩ were included in the analysis. Since the voltage was not changed during the course of these experiments, we did not correct for voltage errors due to uncompensated series resistance.

## Acknowledgements

We thank Z Liao for technical assistance with *C. elegans* animals. Experimental research was supported by NIH grants (R01-NS-047715, R35-NS-105092 to MBG; R01-EB-006745 to BLP and MBG) and fellowships (F31-NS-093825 to SK).

## Additional information

### Funding

| Funder | Grant reference number | Author |
|---|---|---|
| National Institutes of Health | R01-NS-047715 | Miriam B Goodman |
| National Institutes of Health | R35-NS-105092 | Miriam B Goodman |

| National Institutes of Health | R01-EB-006745 | Beth L Pruitt |
| National Institutes of Health | F31-NS-093825 | Samata Katta |

The funders had no role in study design, data collection and interpretation, or the decision to submit the work for publication.

### Author contributions
Alessandro Sanzeni, Conceptualization, Software, Formal analysis, Validation, Investigation, Methodology, Writing—original draft; Samata Katta, Conceptualization, Formal analysis, Validation, Investigation, Methodology, Writing—original draft; Bryan Petzold, Validation, Investigation, Visualization, Writing—original draft; Beth L Pruitt, Supervision, Validation, Investigation, Writing—original draft; Miriam B Goodman, Massimo Vergassola, Conceptualization, Formal analysis, Supervision, Validation, Investigation, Methodology, Writing—original draft

### Author ORCIDs
Alessandro Sanzeni https://orcid.org/0000-0001-8758-1810
Samata Katta http://orcid.org/0000-0001-9748-1653
Beth L Pruitt http://orcid.org/0000-0002-4861-2124
Miriam B Goodman https://orcid.org/0000-0002-5810-1272
Massimo Vergassola https://orcid.org/0000-0002-7212-8244

### Decision letter and Author response
Decision letter https://doi.org/10.7554/eLife.43226.044
Author response https://doi.org/10.7554/eLife.43226.045

## Additional files

### Supplementary files
• Supplementary file 1. Key Resources Table.
DOI: https://doi.org/10.7554/eLife.43226.014

• Transparent reporting form
DOI: https://doi.org/10.7554/eLife.43226.015

### Data availability
All data generated or analysed during this study are included in the manuscript and supporting files. The electrophysiology source data from Eastwood et al. is available on Dryad (https://doi.org/10.5061/dryad.82mn2ht). See also transparent reporting form.

The following previously published dataset was used:

| Author(s) | Year | Dataset title | Dataset URL | Database and Identifier |
|---|---|---|---|---|
| Eastwood AL, Sanzeni A, Petzold BC, Park S, Vergassola M, Pruitt BL, Goodman MB | 2019 | Data from: Tissue mechanics govern the rapidly adapting and symmetrical response to touch | https://doi.org/10.5061/dryad.82mn2ht | Dryad Digital Repository, 10.5061/dryad.82mn2ht |

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

## Appendix 1

DOI: https://doi.org/10.7554/eLife.43226.016

### Numerical evaluation of the amplitude of the deformation gradients

In the main text, we exploited the property that gradients of the deformation field **u** (which are non-dimensional quantities) are small compared to unity. This was used first to justify a Hookean relation between stress and strain, and then in the derivation of the friction force acting on the elastic filaments. The goal of this Appendix is to provide an empirical a posteriori validation of this assumption.

*Appendix 1—figure 1* shows the various gradients components for an indentation of 10 µm, which is the greatest value in experiments. To provide an upper bound, we focus on the region of maximum deflection, that is along the longitudinal direction at the top of the cylinder. Over a wide range of positions, both in soft and stiff worms, the gradients are indeed smaller than unity. The only exception is the component $du_z/dy$, which approaches unity in a small region around $y = 5$ µm, which is where the bead detaches. However, it is sensible to neglect even this contribution for predictions of the neural current. Indeed, values in *Appendix 1—figure 1* are an upper bound, and the region is just a few µm wide, hence only a small number of channels are possibly concerned (we remind that the average inter-channel distance is ~2 µm). Note also that $\frac{du_y}{dz}$ has a negative value that compensates for $\frac{du_z}{dy}$ in the yz component of the strain tensor, which could otherwise potentially be large.

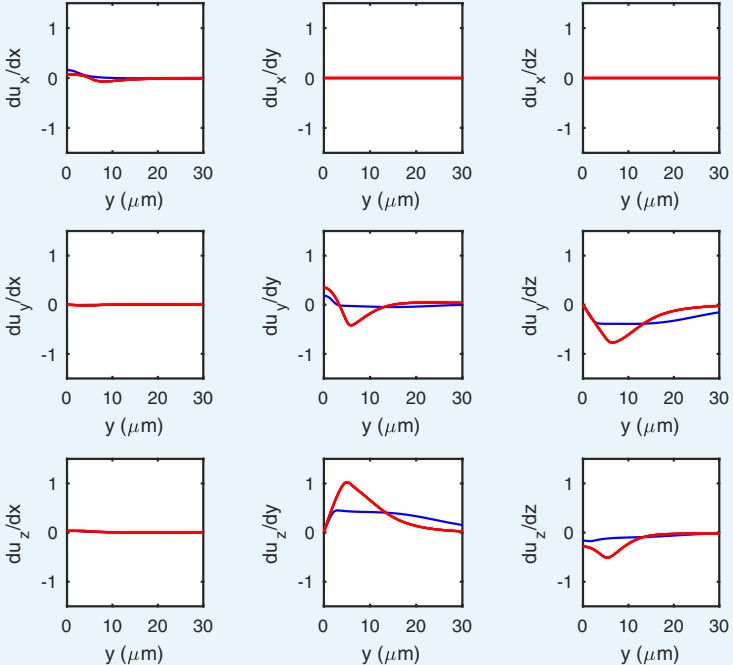

**Appendix 1—figure 1.** The gradients of the deformation along the longitudinal direction of an indented cylinder. Gradients of **u** computed by using numerical simulations for soft ($p/E = 4 \times 10^{-4}$, blue) and stiff ($p/E = 0.01$, red) shells. Note their moderate amplitude even for the indentation of 10 µm shown here, which is the strongest that we consider.

DOI: https://doi.org/10.7554/eLife.43226.017

## Appendix 2

DOI: https://doi.org/10.7554/eLife.43226.016

### How does the radius of the cylinder change with the internal pressure?

In this Appendix we discuss modifications in the geometry of a cylindrical shell upon application of internal pressure. We first compute analytically the deformation field in the linear approximation, and then extend the derivation in the nonlinear regime with the assistance of numerical analysis.

The 3D elasticity equations (**Equation (2)** in the main text) in the linear approximation and cylindrical coordinates read

$$\frac{\partial \sigma_{rr}}{\partial r} + \frac{\sigma_{rr} - \sigma_{\theta\theta}}{r} = 0, \quad \frac{1}{r}\frac{\partial \sigma_{\theta\theta}}{\partial \theta} = 0, \quad \frac{\partial \sigma_{yy}}{\partial y} = 0, \tag{14}$$

where $x = r\cos(\theta)$, $z = r\sin(\theta)$; non-diagonal terms of $\sigma$ vanish due to the symmetry of the problem. Boundary conditions are

$$\sigma_{rr}(r = R_{in}) = -p, \quad \sigma_{rr}(r = R_{out}) = 0, \quad \sigma_{yy}(y = \pm L/2) = 0, \tag{15}$$

where $R_{in} = R - t/2$ and $R_{out} = R + t/2$. The condition of zero longitudinal stress at the two ends of the cylinder is motivated by the results in Appendix 7.

The third line of **Equation (14)** and the boundary conditions imply $\sigma_{yy} = 0$ along the shell. By using the constitutive Hookean relations between stress and strain tensor in the main text, we obtain

$$\sigma_{rr} = \frac{E}{1 - \nu^2}(\epsilon_{rr} + \nu\epsilon_{\theta\theta}), \quad \sigma_{\theta\theta} = \frac{E}{1 - \nu^2}(\epsilon_{\theta\theta} + \nu\epsilon_{rr}). \tag{16}$$

For small deformations, the diagonal components of the strain tensor (see **Equation (1)** in the main text) are given by

$$\epsilon_{rr} = \frac{\partial u_r}{\partial r}, \quad \epsilon_{\theta\theta} = \frac{1}{r}\frac{\partial u_\theta}{\partial \theta} + \frac{u_r}{r}, \quad \epsilon_{yy} = \frac{\partial u_y}{\partial y}. \tag{17}$$

Due to the geometry of the problem $\partial u_\theta / \partial \theta = 0$; by using **Equations (14), (16) and (17)** we find the following equation for the radial deformation

$$\frac{d^2 u_r}{dr^2} + \frac{1}{r}\frac{du_r}{dr} - \frac{u_r}{r^2} = 0, \tag{18}$$

whose general solution is $u_r = a/r + br$. Using the boundary conditions we obtain

$$u_r = \frac{p/E}{R_{out}^2/R_{in}^2 - 1}\left((1 + \nu)\frac{R_{out}^2}{r} + (1 - \nu)r\right), \quad u_y = -\nu\frac{p/E}{R_{out}^2/R_{in}^2 - 1}y, \quad u_\theta = 0, \tag{19}$$

$$\sigma_{rr}(r) = \frac{p}{R_{out}^2/R_{in}^2 - 1}\left(1 - \frac{R_{out}^2}{r^2}\right), \quad \sigma_{\theta\theta}(r) = \frac{p}{R_{out}^2/R_{in}^2 - 1}\left(1 + \frac{R_{out}^2}{r^2}\right). \tag{20}$$

It follows from the solution **Equation (19)** that the relative change in radius, thickness and length, in the limit $t/R \ll 1$, are given by

$$\frac{\Delta R}{R} = \frac{pR}{Et}\left(1 + (\nu - 1)\frac{t}{2R}\right), \quad \frac{\Delta t}{t} = -\frac{pR}{Et}\left(\nu + (1 - \nu)\frac{t}{2R}\right), \quad \frac{\Delta L}{L} = -\frac{\nu}{2}\frac{pR}{Et}. \tag{21}$$

**Appendix 2—figure 1** demonstrates good agreement of these predictions with numerical simulations for small values of $p/E$. The corresponding expressions for the change in the volume $V = \pi L(R + t/2)^2$ of the cylinder and its bulk modulus read

$$\frac{\Delta V}{V} \simeq (2 - \nu/2)\frac{pR}{Et}, \quad k_{linear} \simeq E\frac{t}{R(2 - \nu/2)}. \tag{22}$$

As $p/E$ increases, nonlinear behaviors beyond the linear description of *Equations (14) and (17)* become important. An analytical description of the corresponding deformations would require to take into account nonlinear terms in the original *Equations (1) and (2)* of the main text.

For our purpose here, the following empirical approach will suffice. The linear solutions *Equation (21)* depend on two dimensionless small parameters: $pR/Et$ and $t/R$. In fact, *Equation (21)* has $t/R$ appearing only multiplied by $pR/Et$; that makes its contribution small, and implies that the functions depends on $pR/Et$ only, at the dominant order. *Appendix 2— figure 1* shows that this property extends in the nonlinear regime: indeed, the curves for relevant values of $t/R$ collapse when plotted vs $pR/Et$. Using this empirical observation, we looked for a power series in $pR/Et$ and found numerically that the following functional forms describe the deformations in the regime relevant for our problem ($pR/Et \leq 0.4$):

$$\frac{\Delta R}{R} = \frac{pR}{Et}\left(1 + \alpha_R\frac{pR}{Et}\right), \quad \frac{\Delta t}{t} = \frac{pR}{Et}\left(-\nu + \alpha_t\frac{pR}{Et}\right), \quad \frac{\Delta L}{L} = \frac{pR}{Et}\left(-\frac{\nu}{2} + \alpha_L\frac{pR}{Et}\right), \tag{23}$$

where $\alpha_R = -0.6182$, $\alpha_t = -0.0626$, $\alpha_L = 0.0479$.

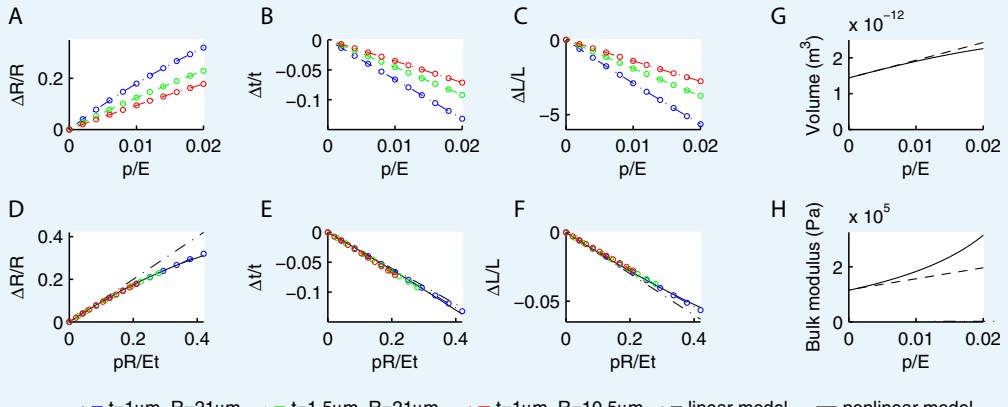

**Appendix 2—figure 1.** Deformations of a cylindrical shell due to internal pressure. Relative change in the radius $R$ (**A**), thickness $t$ (**B**) and length $L$ (**C**) of the shell as a function of $p/E$ and various $t/R$, as obtained from our numerical simulations. As $p$ increases, $R$ increases whilst $t$ and $L$ decrease. (**D,E,F**) While the curves in the previous panels change with $t/R$, the curves are collapsed by plotting them against $pR/Et$, which suggests that the contribution of terms in $t/R$ is negligible. The collapsed behavior agrees with the linear prediction *Equation (21)* for $pR/Et<0.1$ (dash-dotted lines), and is well captured by the empirical *Equation (23)* in the moderately nonlinear regime. Using the nonlinear *Equation (23)*, we computed the change in volume (**G**) and bulk modulus (**H**) of the shell as a function of $p/E$ (dashed lines are the linear predictions *Equation (22)* valid for small values of $p/E$).
DOI: https://doi.org/10.7554/eLife.43226.019

Finally, by using *Equation (23)*, we obtain the volume $V$ and the bulk modulus $k$ of the cylinder as a function of $p/E$, as shown in *Appendix 2—figure 1G,H*.

# Appendix 3

DOI: https://doi.org/10.7554/eLife.43226.016

## Effects of external atmospheric pressure on mechanical and neural response

In the main text, we studied the mechanical properties of *C. elegans* body as a function of the pressure parameter $p$, that is the difference between the internal and the external atmospheric pressure $P_{atm}$, neglecting the latter. This Appendix shows that our results hold also when $P_{atm}$ is taken into account.

To gain insight, we first adapted the calculation in Appendix 2 to the case where atmospheric pressure $P_{atm}$ is considered. The equations are not modified yet the boundary conditions on the internal and external surface of the shell become

$$P_{in} = P_{atm} + p, \quad P_{out} = P_{atm}. \tag{24}$$

By following the same procedure detailed in the previous Section B, we obtain that the radial deformation of a pressurized shells is given by

$$\frac{\Delta R}{R} \simeq \frac{pR}{Et}\left[1 + (\nu - 1)\frac{t}{2R}\left(1 + \frac{2P_{atm}}{p}\right)\right]. \tag{25}$$

*Equation (25)* suggests that effects of atmospheric pressure on the mechanical response of the shell are negligible as long as $P_{atm}t/pR \ll 1$. To test this hypothesis, we simulated indentation experiments with and without atmospheric pressure for different values of $p$; results are shown in *Appendix 3—figure 1*. As expected, for $p/E \approx 10^{-2}$, both the force indentation relation and the deformation profile are not modified by the atmospheric pressure. It follows that all mechanical properties and neural responses derived in the main text for stiff worms (where the internal pressure is high and $p/E \approx 10^{-2}$) are not modified by the inclusion of atmospheric pressure.

Soft worms were shown in the main text to have smaller values of $p/E$ because of the dissection procedure. *Appendix 3—figure 1* shows that the force indentation relation does not change significantly yet the deformation profile becomes wider as $p/E$ reduces. This modification is not relevant to describe mechanical properties of intact animals but it might *a priori* influence the neural response in soft worms. Therefore, we computed numerically the neural response to indentation experiments in soft worms with and without atmospheric pressure. As shown in *Appendix 3—figure 1*, the performance of the model in describing neural data is not modified substantially, which shows that results of the main text are generally valid even when atmospheric pressure is taken into account.

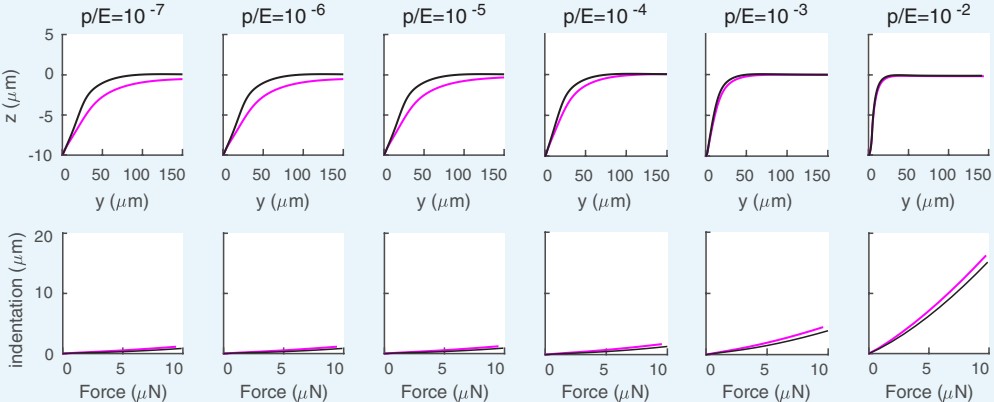

**Appendix 3—figure 1.** Effect of atmospheric pressure on mechanical response. Deformation

profile (first row) and force indentation relation (second row) for a cylindrical shell (with the same properties as in the main text) with (purple) and without (black) atmospheric pressure.
DOI: https://doi.org/10.7554/eLife.43226.021

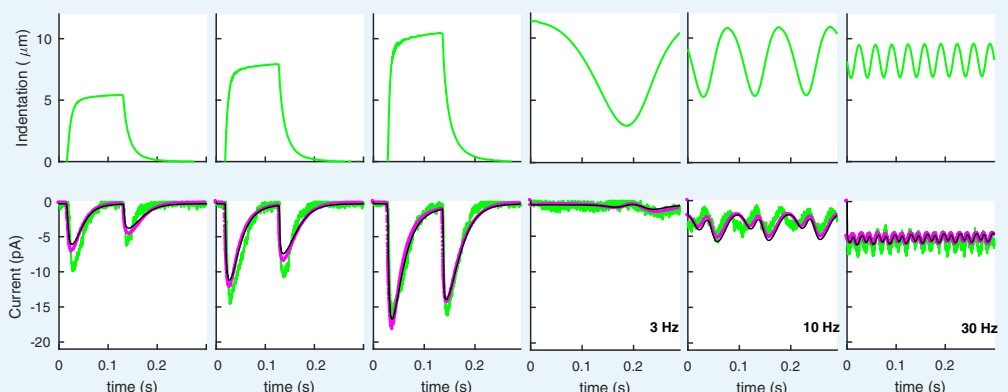

**Appendix 3—figure 2.** Effect of atmospheric pressure on neural response. Neural responses of the model presented in the main text to the experimental stimuli in *Figure 3* of the main text. Green curves are experimental data (as in *Figure 3* of the main text); purple and black curves are model predictions with and without atmospheric pressure, respectively. The ratio $p/E$ is 7‰ of the value for stiff worms.
DOI: https://doi.org/10.7554/eLife.43226.022

## Appendix 4

DOI: https://doi.org/10.7554/eLife.43226.016

### The thin shell limit

In the limit when the shell is thin and the surface is shallow, the 3D elasticity *Equation (2)* reduce to (*Ventsel and Krauthammer, 2001*; *Audoly and Pomeau, 2011*):

$$\begin{cases} B\nabla^4 u_z + \nabla_k^2 \phi - [\phi, u_z] = p - F(x,y), \\ \frac{1}{S}\nabla^4 \phi - \nabla_k^2 u_z = -\frac{1}{2}[u_z, u_z], \end{cases} \tag{26}$$

which is the limit equation that was used in the main text for analyzing the energetic balance between bending, stretching and internal pressure. The brackets in *Equation (26)* are defined as $[f,g] \equiv \frac{\partial^2 f}{\partial x^2}\frac{\partial^2 g}{\partial y^2} - 2\frac{\partial^2 f}{\partial x \partial y}\frac{\partial^2 g}{\partial x \partial y} + \frac{\partial^2 g}{\partial x^2}\frac{\partial^2 f}{\partial y^2}$, while the derivative $\nabla_k^2 = \frac{1}{R}\frac{\partial^2}{\partial y^2}$. *Equation (26)* is two-dimensional, with the variables $z = z(x,y)$ and $u_z = u_z(x,y)$ representing the middle surface and the deformation field of the cylindrical shell, respectively. The deformed surface is $z + u_z$ and we chose the axes so that the plane $z = 0$ is tangent to the top of the cylinder. The Airy stress function $\phi$ is the scalar function that parametrizes the in-plane components of the stress tensor as $\sigma_{xx} = \partial_{yy}^2 \phi$, $\sigma_{yy} = \partial_{xx}^2 \phi$ and $\sigma_{xy} = -\partial_{xy}^2 \phi$.

The above simplifications are due to the thinness of the shell and the resulting small vertical components of $\sigma$. The parameters $B = Et^3/12(1-\nu^2)$ and $S = Et$ are the bending and stretching stiffness, respectively. Finally, $p$ and $F$ are the internal pressure and the external force applied by the indenter. In the limit $R \to \infty$, the $\nabla_k^2$ term is negligible, and *Equation (26)* reduce to the Föppl-von Karman equations for a thin plate (*Ventsel and Krauthammer, 2001*; *Audoly and Pomeau, 2011*).

# Appendix 5

DOI: https://doi.org/10.7554/eLife.43226.016

## Validation of the numerical scheme used to determine the mechanical response

In the main text, we described the mechanics of the nematode as an elastic cylindrical shell under pressure. Because of the geometrical nonlinearities involved, numerical simulations are the main tool available for determining the resulting mechanical response. The goal of this Appendix is to give more details on the tests that we employed to validate the numerical procedure discussed in the main text. Tests rely on elasticity problems that were previously investigated, namely small indentations of cylindrical shells (where an analytical solution is available [**Morley, 1960**]), and large indentations of pressurized spherical shells (where a simplified framework was derived in **Vella et al., 2012a**). In all cases, agreement was obtained.

### 1 Small indentations of cylindrical shells

A thin cylindrical shell subject to equal and opposite concentrated radial loads was investigated in **Morley (1960)**. For indentations $w_0 \ll t$, where $t$ is the shell thickness, the equations of 3D elasticity (**Equation (2)** in the main text) reduce to (**Morley, 1959**):

$$\nabla^4 \left( \nabla^2 + \frac{1}{R^2} \right)^2 w + \frac{4K^4}{R^4} w_{,xxxx} = \frac{1}{D} \nabla^4 q \,, \tag{27}$$

where $q$ is the applied load, $x$ and $\theta$ are the cylindrical longitudinal and angular coordinates, $w$ is the radial displacement and

$$K^4 = 3(1 - \nu^2)\left(\frac{R}{t}\right)^2, \quad D = \frac{Et^3}{12(1-\nu^2)} \,,$$
$$w_{,xxx\theta} = \frac{1}{R}\frac{\partial^4 w}{\partial x^3 \partial \theta}, \quad w_{,xxxx} = \frac{\partial^4 w}{\partial x^4}, \quad \nabla^2 = \frac{\partial^2}{\partial x^2} + \frac{1}{R^2}\frac{\partial^2}{\partial \theta^2} \,. \tag{28}$$

By 'equal and opposite concentrated radial loads', it is meant that two equal, spatially localized forces are applied at $(x, \theta) = (0, 0)$ and $(x, \theta) = (0, \pi)$.

The solution to the above problem was obtained (**Morley, 1960**) by writing $q(x, \theta)$ as

$$q(x, \theta) = \frac{F}{\pi^2 R^2} \lim_{\delta \to 0} \int_0^\infty \left[ \frac{\sin(\lambda\delta/R)}{\lambda\delta/R} + 2 \sum_{n=2,4,\ldots}^\infty \frac{\sin(n\delta/R)\sin(\lambda\delta/R)}{(n\delta/R)(\lambda\delta/R)}\cos(n\theta) \right] \cos(\lambda x/R) d\lambda \,, \tag{29}$$

and solving **Equation (27)** order by order in $n$. The resulting deformation profile reads (**Morley, 1960**) :

$$w(x, \theta) = \frac{R^2 F}{2\pi D} \Big\{ \frac{e^{-xK_-}}{4K^4} [RK_+ \cos(xK_+) + RK_- \sin(xK_+)] -$$
$$-2 \sum_{n=2,4,\ldots}^\infty \mathrm{Im} \left[ \frac{\lambda_{1n}(\lambda_{1n}^2 + n^2)e^{i\lambda_{1n}x/R}}{(\lambda_{1n}^2 + n^2 - 1)(\lambda_{1n}^4 - n^4 + n^2)} + \frac{\lambda_{2n}(\lambda_{2n}^2 + n^2)e^{i\lambda_{2n}x/R}}{(\lambda_{2n}^2 + n^2 - 1)(\lambda_{2n}^4 - n^4 + n^2)} \right] \cos(n\theta) \Big\} \,, \tag{30}$$

with

$$K_\pm = \frac{1}{R}\sqrt{K^2 \pm \frac{1}{2}}, \quad \lambda_{1n,2n}^2 = -n^2 + \frac{1}{2} \mp iK^2 \left[ 1 \mp \sqrt{1 \mp \frac{i(2n^2 - 1)}{K^2}} \right] . \tag{31}$$

The force-indentation relation is obtained from **Equation (30)** by determining the deformation $w$ at either one of the loading points, that is $w(0,0)$ in the above formulæ, as a function of $F$. An example of the force-indentation relation and the deformation profile predicted by **Equation (30)** are shown in **Appendix 5—figure 1**.

By using the numerical approach described in the main text, we determined the mechanical response of a thin cylindrical shell, and compared it to the above solution. Note that we are solving directly the equations of three-dimensional elasticity, at variance with the simplified set of equations in *Morley (1960)*. Results for different mesh sizes are shown in *Appendix 5—figure 1*: the deformation profile is indeed captured by our code, even with a relatively coarse mesh; a finer mesh is needed to capture quantitatively the force-indentation relation.

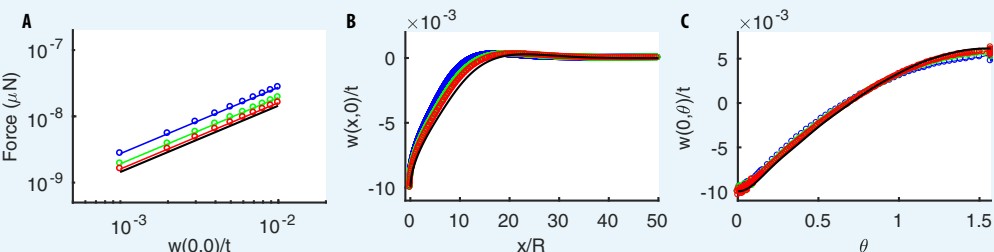

**Appendix 5—figure 1.** Mechanical response of a thin cylindrical shell to equal and opposite concentrated radial loads. (**A**) Force-indentation relation. (**B–C**) Radial deflection along the longitudinal and angular directions. The analytical solution *Equation (30)* (black line) is well approximated by numerical solutions (colored lines) obtained by using our numerical code. The agreement improves as the numerical mesh becomes finer (the mesh length is $t$ (blue), $t/2$ (green), $t/3$ (red), where $t$ is the thickness of the shell). Parameters of the simulations are: $t/R = 10^{-2}$, $t = 1\mu$m, $E = 1$ MPa, $\nu = 0.3$.
DOI: https://doi.org/10.7554/eLife.43226.025

## 2 Large indentation of spherical shells

Large indentations of pressurized spherical shallow shells were investigated in *Vella et al. (2012a)*. In response to a point indentation (and in the absence of buckling [*Vaziri and Mahadevan, 2008*]), the 3D equations of nonlinear elasticity reduce to

$$\begin{cases} \frac{F}{2\pi} = \frac{pr^2}{2} + \psi\left(\frac{dw}{dr} - \frac{r}{R}\right), \\ r\frac{d}{dr}\left[\frac{1}{r}\frac{d}{dr}(r\psi)\right] = Et\left[\frac{r}{R}\frac{dw}{dr} - \frac{1}{2}\left(\frac{dw}{dr}\right)^2\right]. \end{cases} \quad (32)$$

Here, $r$ indicates the distance in the plane orthogonal to the indentation point, $p$ is the internal pressure, $w(r)$ is the deformation field, and $\psi$ is related to the components of the stress tensor as $\sigma_{\theta\theta} = \psi'$ and $\sigma_{rr} = \psi/r$. The nonlinear term in *Equation (32)* is due to geometrical nonlinearities generated by large deformations. Boundary conditions are:

$$w(0) = -w_0, \quad \lim_{r \to 0}(r\psi' - \nu\psi) = 0, \quad w(\infty) = 0, \quad \psi'(\infty) = \frac{pR}{2}, \quad (33)$$

where the prime denotes differentiation with respect to $r$.

A test of our numerical scheme is provided by the comparison of its results for a thin spherical shell to the solution of the simplified *Equation(32)*. Examples of force-indentation relations and deformation profiles given by *Equation(32)* for different values of $p$ are shown in *Appendix 5—figure 2*. Results of our code are also compared in *Appendix 5—figure 2*: both the force-indentation relation and the deformation profile are well reproduced.

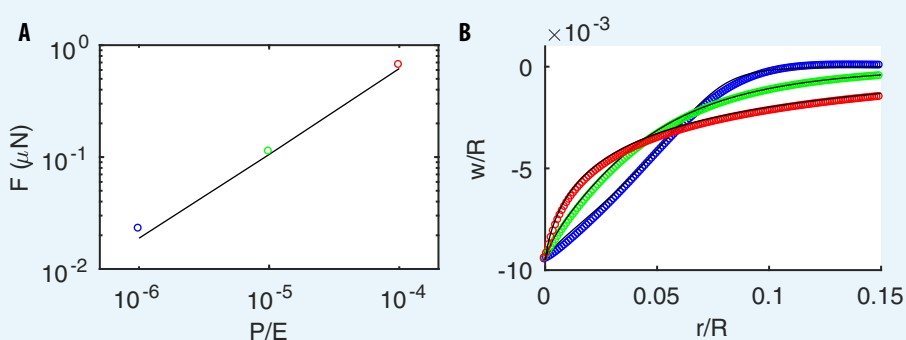

**Appendix 5—figure 2.** Mechanical response of a pressurized thin spherical shell to a point indentation at the north pole. (**A**) The force required to produce a deformation $w_0/R = -10^{-2}$ for different values of $p$. (**B**) The deformation profile for $p/E = 10^{-6}$ (blue), $10^{-5}$ (green), $10^{-4}$ (red). Black lines and colored dots correspond to the solutions of **Equation (32)** and the results by our code, respectively. Note that, as for a pressurized cylinder (see main text), the deformation profile narrows as $p$ increases. Parameters of the simulations are : $E = 1\,\text{MPa}$, $\nu = 0.3$, $t/R = 10^{-3}$, $t = 1\,\mu\text{m}$.

DOI: https://doi.org/10.7554/eLife.43226.026

# Appendix 6

DOI: https://doi.org/10.7554/eLife.43226.016

## How the gluing of the nematode onto the plate influences its mechanical response

Standard touch sensation experiments have the nematode glued onto a plate. As mentioned in the main text, the displacement of its body is strongly limited at locations where the glue is applied. This Appendix will analyze the effect of the gluing onto mechanical responses.

We consider the limiting case where only the line of contact with the plate is glued, that is the limit opposite to the one considered in the main text. There, the entire lower half of the body was glued, which was motivated by the experiments reported in the paper. In our model, the south pole of the cylinder corresponds to the line of contact with the plate in the absence of indentation. *Appendix 6—figure 1* shows the corresponding response of the shell to indentation.

It is of interest that the stiffness in *Appendix 6—figure 1A* is smaller for the south-pole gluing, even though its longitudinal deformation is more extended than for the lower-half gluing, as visible in *Appendix 6—figure 1B*. That is accounted by the deformation along the orthogonal direction, which expands in the entire lower half of the body (see *Appendix 6—figure 1C*). That deformation is forbidden for the lower-half gluing, which is the reason for the increased stiffness. Generally, the stiffness is expected to decrease as we reduce the region where the nematode is glued to the plate, which could be verified experimentally.

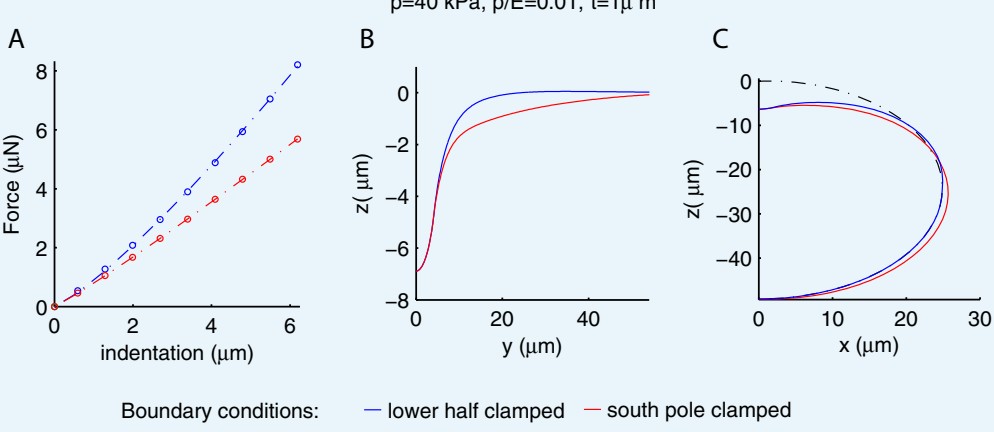

**Appendix 6—figure 1.** Influence of the gluing of the worm on its mechanical response. Blue and red curves correspond respectively to gluing of the entire lower half of the cylinder or the line of contact with the plate (south pole) only. (**A**) Force-indentation relations; note that the stiffness is greater for the blue curve. The deformation profiles along the longitudinal (**B**) and orthogonal (**C**) directions are wider for the south-pole gluing, which has the lower half of the shell deformed in the orthogonal direction as well. The undeformed geometry is the black dash-dotted line.

DOI: https://doi.org/10.7554/eLife.43226.028

# Appendix 7

DOI: https://doi.org/10.7554/eLife.43226.016

## Influence of the boundary conditions at the ends of the cylinder on its mechanical response

The main text discusses a pressurized cylindrical shell with free conditions at the two ends of the cylinder. Here, we analyze how mechanical properties are modified when the ends are closed by plugs, which leads to an additional component of stress.

We computed numerically the response to the indentation of a closed pressurized cylindrical shell. The numerical procedure is quite analogous to the main text, with the only difference of the boundary conditions. The action of the pressure on the plugs produces a longitudinal force on the shell, whose magnitude does not depend on their structure (but for a boundary layer close to the ends). Without loss of generality, we used semi-spherical plugs.

Results of the simulations are shown in **Appendix 7—figure 1**. The deformation profile for a given value of $p/E$ is more extended than for free conditions at the sides (**Appendix 7—figure 1A**). Since the associated change in volume is bigger, and the stiffness is dominated by internal pressure, the stiffness of the shell is greater (**Appendix 7—figure 1B**). The additional stiffness stems from the longitudinal stress introduced by the lateral plugs, as confirmed by the decrease of the difference between the closed and the open conditions with the internal pressure (**Appendix 7—figure 1A,B**).

Let us now show that closed ends cannot reproduce experimental data, which is the reason why the main text focuses on open cylinders. We first fix the range $p/E \in [0, 0.02]$ considered so far. The extension of the deformation decreases with $p/E$ but, even at the largest value $p/E = 0.02$, it remains too wide to account for experimental data. Further increase in $p/E$ does reduce the extension, yet it runs in conflict with experimental data on the bulk modulus (**Gilpin et al., 2015**). Indeed, the estimate for $p$ obtained in the main text depends only on the deformation profile and the force-indentation relation, which are both given by the experiments. We can therefore fix $p = 40k$Pa, and predict the bulk modulus for the corresponding various values of $E$. Results in **Appendix 7—figure 1C** are systematically smaller than experiments (and even further increase of $p/E$ would not help as the bulk modulus decreases with $p/E$).

In summary, we showed that the description of the nematode body as an elastic shell with closed ends cannot reproduce experimental data on the mechanics of *C. elegans*. Conversely, the main text showed that the same elastic model with free sides does capture main features of the mechanical response. Our results suggest that the longitudinal stress generated by the plugs is somehow relaxed in the worm, which may relate to the annular structure of the cuticle.

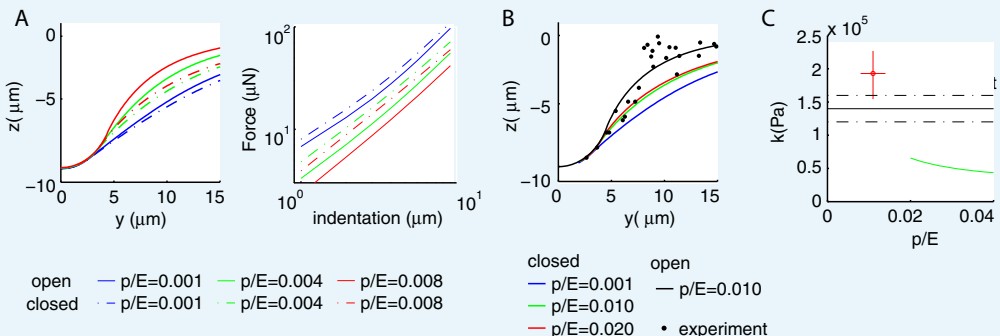

**Appendix 7—figure 1.** Boundary conditions at the ends of the shell influence mechanical properties. (**A**) Comparison of the mechanical response for a closed (dash-dot line) and open (continuous line) cylinder. For a given value of $p/E$, the deformation profile is more

extended (left) and the shell stiffer (right) if the two ends are closed; the difference increases with $p/E$. (**B**) Experimental and numerical deformation profiles along the longitudinal axis. None of the values $p/E \in [0, 0.02]$ for the closed cylinder (colored line) captures experimental data. (**C**) Experimental and predicted values for the bulk modulus. The value predicted for closed conditions at the ends decreases with $p/E$ and is too small to account for the data. Results for free lateral conditions are shown for comparison. Parameters of the simulations are as in *Figure 2* of the main text.

DOI: https://doi.org/10.7554/eLife.43226.030

## Appendix 8

DOI: https://doi.org/10.7554/eLife.43226.016

### The local geometry of the channels along the TRNs

We treat the TRN as a (small) cylinder running (at rest) in the upper part of the (big) cylinder in **Figure 1**, at $x = 0$ along the $y$ axis. The TRN is placed right above the internal part of the shell. Upon indentation, the orthogonal basis $\hat{\mathbf{x}}_i$ ($i = 1, 2, 3$) in **Figure 1** is deformed into a triad of vectors $\hat{\mathbf{x}}_i'$ in a way that depends on its original location. The separation between pairs of neighboring material points $(\mathbf{r}, \mathbf{r} + \mathbf{dr})$ is calculated using the gradients of the displacement field $\mathbf{u}(\mathbf{r})$. In particular, the variation of the squared distance $dr'^2 - dr^2 = 2\varepsilon_{ij} dr_i \, dr_j$, and the angle between two vectors $\mathbf{dr_1}$ and $\mathbf{dr_2}$ (see **Landau et al., 1986**; **Audoly and Pomeau, 2011**)

$$\mathbf{dr}_1' \cdot \mathbf{dr}_2' - \mathbf{dr}_1 \cdot \mathbf{dr}_2 = 2\varepsilon_{ij} dr_{1,i} \, dr_{2,j}. \tag{34}$$

A convenient orthonormal basis $\hat{\mathbf{e}}_i'$ to analyze the dynamics of the channels is defined as follows : $\hat{\mathbf{e}}_y'$ is aligned with the local direction of the (deformed) axis of the cylinder running head-to-tail; $\hat{\mathbf{e}}_z'$ is orthogonal to the neural membrane at the top of the TRN, and oriented outward; $\hat{\mathbf{e}}_x'$ is tangential to the neural membrane, along the remaining direction of a right-handed system.

For every channel, we define a local base $\hat{\mathbf{w}}_i$ such that $\hat{\mathbf{w}}_{1,2}$ span the plane locally tangential to the neural membrane while $\hat{\mathbf{w}}_3$ indicates the orthogonal direction. The bases $\hat{\mathbf{w}}_i$ are constructed by rotating the $\hat{\mathbf{e}}_i'$ appropriately. For a channel placed at the top of the TRN, the local basis $\hat{\mathbf{w}}_i$ coincides with $\hat{\mathbf{e}}_i'$. If the channel is rotated by $\theta$ along the surface of the TRN, then $\hat{\mathbf{w}}_1 = \cos(\theta)\hat{\mathbf{e}}_x' - \sin(\theta)\hat{\mathbf{e}}_z'$, $\hat{\mathbf{w}}_2 = \hat{\mathbf{e}}_y'$, and $\hat{\mathbf{w}}_3 = \sin(\theta)\hat{\mathbf{e}}_x' + \cos(\theta)\hat{\mathbf{e}}_z'$.

# Appendix 9

DOI: https://doi.org/10.7554/eLife.43226.016

## Comparison of different functional forms for the activation of the channels

In our model, the free energy of a channel is modulated by the deformation of its elastic filament. A general rotationally-symmetric form of the free energy reads

$$\beta\Delta G_{oc}(\mathbf{x}) = g_0 + g_1 x + g_2 x^2 \ldots \tag{35}$$

where $x = |\mathbf{x}|$. In the main text, we used the linear form of *Equation (35)*; here, we repeat the analysis of the experimental data for $g_1 = 0$, that is a quadratic form. Results are shown in *Appendix 9—figure 1*, with results for the linear model included for the sake of comparison. The quadratic dependence limits the sensitivity range, and leads to a worse description of the data (see *Appendix 9—figure 1*); the same also holds for individual realizations of the responses (data not shown). That is witnessed by the step response in *Appendix 9—figure 1*, where the current predicted by the model saturates at smaller values than the peaks observed in the data, even though the best-fitted baseline activity is stronger.

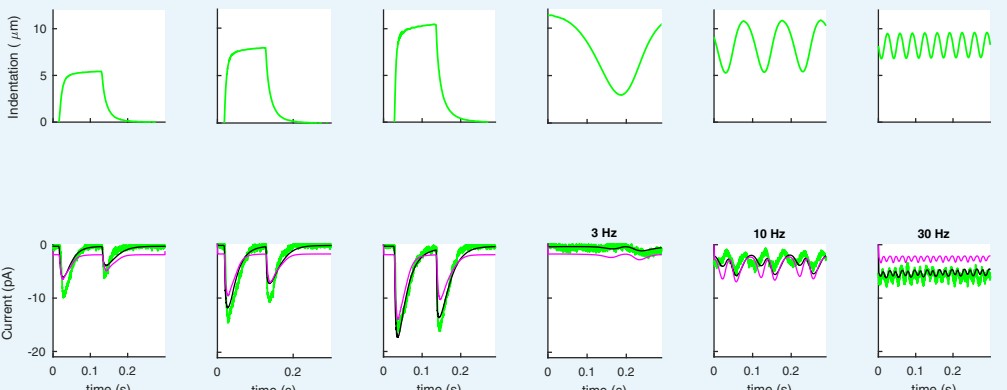

**Appendix 9—figure 1.** The linear form of *Equation (35)* outperforms its quadratic counterpart in describing experimental data. (**A**) The average predictions for the experimental data (green) given by the linear (black) and quadratic (purple) forms of *Equation (35)*.
DOI: https://doi.org/10.7554/eLife.43226.033

We also repeated the analysis of the directional model replacing *Equation (13)* of the main text by

$$\beta\Delta G_{oc} = g_0 - g_2\,\mathcal{F}\cdot\mathbf{v}\,\Theta(\mathcal{F}\cdot\mathbf{v}), \tag{36}$$

where $\Theta$ is the Heaviside function. Results were comparable to those presented in the main text.

## Appendix 10

DOI: https://doi.org/10.7554/eLife.43226.016

## Noise analysis for non-equally stimulated channels

Here, we compute mean and variance of the current as a function of the statistics of the ion channels. We use these results in the main text to compare model predictions to experimental data. The point is to generalize standard noise analysis to the case where the channels are not equivalent. That can be the case either because they are not identical or, as it the case here, because their stimulation differs due to their location with respect to the stimulation. We also calculate the scaling of the level of fluctuations expected in a finite sample of $N$ measurements.

We start by deriving the expression of the mean and variance of the neural current from the statistics of the single channels. The total ion current $I$ through the neuron is the sum $I = \sum_{k=1}^{K} i_k$, where $i_k$ is the current flowing through the $k$-th channel and the sum runs over the channels along the neuron. Channels can take three distinct states in our model : open, with maximal current $i_0$ and probability $P_o(k)$; sub-conducting, with intermediate current $i_s$ and probability $P_s(k)$; closed, with no current and probability $P_c(k) = 1 - P_o(k) - P_s(k)$. Each channel follows a generalized Bernoulli distribution : the associated mean and variance are

$$\langle i_k \rangle = i_o P_o(k) + i_s P_s(k), \quad \sigma_k^2 = i_o^2 P_o(k)(1 - P_o(k)) + i_s^2 P_s(k)(1 - P_s(k)) - 2i_o i_s P_o(k) P_s(k). \quad (37)$$

Assuming that the gatings of the channels are independent random variables, we obtain

$$\langle I \rangle = \sum_{k=1}^{K} \langle i_k \rangle, \quad \sigma_I^2 = \langle (I - \langle I \rangle)^2 \rangle = \sum_{k=1}^{K} \sigma_k^2. \quad (38)$$

In the main text we suggested that the variance of the current could be used to study experimentally the microscopic properties of ionic channels. Since the variance should be inferred from a finite sample, we want to quantify the scaling of fluctuations in the sample variance with the number of measurements. Given $N$ measurements $I_n$ of the TRN current, the sample mean $m_1$ and sample variance $m_2$ are defined as

$$m_1 = \frac{1}{N} \sum_{n=1}^{N} I_n, \quad m_2 = \frac{1}{N} \sum_{n=1}^{N} (I_n - m_1)^2. \quad (39)$$

The expected sample variance and its variance are (**Kenney and Keeping, 1951**; **Rose and Smith, 2002**):

$$\langle m_2 \rangle = \frac{N-1}{N} \mu_2(I), \quad \langle var(m_2) \rangle = \frac{(N-1)^2}{N^3} \mu_4(I) - \frac{(N-1)(N-3)}{N^3} \mu_2(I)^2, \quad (40)$$

$$\mu_2(I) = \langle (I - \langle I \rangle^2 \rangle = \sigma_I^2, \quad \mu_4(I) = \langle (I - \langle I \rangle^4 \rangle. \quad (41)$$

To determine the relation between the moments $\mu_2(I)$, $\mu_4(I)$, ... of the total current $I$ and the statistics of the single-channel currents, it is convenient to use cumulants and the cumulant-generating function of $I$, defined as $Q_I(t) = \log\langle \exp(tI) \rangle$ (**Papoulis, 1991**). Its advantage is that the function is additive :

$$Q_I(t) = \log\langle e^{tI} \rangle = \log\langle e^{t \sum_k i_k} \rangle = \sum_k \log\langle e^{ti_k} \rangle = \sum_k Q_{i_k}(t), \quad (42)$$

where we have exploited independence among the $i_k$'s. Since the cumulant $q_n(I)$ of order $n$ (see **Papoulis, 1991**) is $q_n(I) \equiv \frac{d^n}{dt^n} Q_I(t)\big|_{t=0}$, **Equation (42)** implies the additivity $q_n(I) = \sum_k q_n(i_k)$ of the cumulants.

The cumulants $q_n(i_k)$ are calculated using the fact that the channels obey a generalized Bernoulli distribution :

$$Q_{i_k}(t) = \log\left[(1 - P_s(k) - P_o(k)) + P_s(k)\,e^{ti_s} + P_o(k)\,e^{ti_0}\right], \tag{43}$$

whence we obtain

$$
\begin{aligned}
q_1(i_k) &= i_s P_s(k) + i_o P_o(k)\,, \\
q_2(i_k) &= i_s^2 P_s(k)(1 - P_s(k)) + i_o^2 P_o(k)(1 - P_o(k)) - 2 i_s i_o P_s(k) P_o(k)\,, \\
q_4(i_k) &= i_s^3[i_s - 4q_1(i_k)]P_s(k) + i_o^3[i_o - 4q_1(i_k)]P_o(k) + 6(q_1(i_k))^2 q_2(i_k) + 3(q_1(i_k))^4 - 3(q_2(i_k))^2\,.
\end{aligned}
\tag{44}
$$

The derivation is completed by relating the central moments $\mu(I)$ to its cumulants via standard formulæ (**Papoulis, 1991**), for example

$$\mu_2(I) = q_2(I)\,, \quad \mu_4(I) = q_4(I) + 3(q_2(I))^2\,,\ldots\,, \tag{45}$$

by expressing $q_n(I)$ as $\sum_k q_n(i_k)$, and finally using **Equation (44)**.

## Appendix 11

DOI: https://doi.org/10.7554/eLife.43226.016

### Mechanical effects of mutations in proteins of the cuticle

Mutations of proteins composing the cuticle have been used as a probe to investigate mechanical properties of *C. elegans* (*Park et al., 2007*; *Gilpin et al., 2015*). This Appendix will explore the consequences of the simplest possible assumptions on the effects of those mutations, obtain predictions for our mechanical model, and compare them to experiments.

We shall describe the effects of mutations in the cuticle through variations in the stretching stiffness $S$, which is the only parameter related to the thin external layers (see Appendix 4). We assume that mutations do not affect the internal pressure of the nematode. The argument in *Park et al. (2007)* is that the genes involved, for example *dpy-5* and *lon-2*, do not affect transport proteins likely to regulate osmotic pressure. It is possible, though, that mutations in the cuticle affect the development of the nematode, hence its body structure and the internal pressure. To constructively advance the issue, we assume that that does not happen and explore consequences.

*Appendix 11—figure 1A* shows the dependence of geometrical properties on $S$ : the radius $R$ of the pressurized cylinder decreases, whilst its length $L$ increases with $S$. This reflects changes with respect to the unpressurized condition brought by the internal pressure $p$. *Appendix 11—figure 1B* shows that the bulk modulus increases with $S$. Finally, the stiffness $S$ enters the response to indentation experiments. *Appendix 11—figure 1B* shows the ratio $f \equiv F/w_0$, that is the force $F$ needed to reach an indentation $w_0 = 5\mu m$, vs $S$ (we verified that results do not depend on the choice of $w_0$). The increase with $S$ is due to two contributions that affect the change in volume via the deformation field. First, if the external radius of the shell is kept constant, the deformation field is wider (see *Appendix 11—figure 1B*). Second, the radius of the shell becomes smaller as $S$ increases (see *Appendix 11—figure 1A*) which leads to a larger deformation field (data not shown).

The parameter $S$ is not measured experimentally, which forces us to use $R$ as a proxy. In this formulation, the model predicts that mutations in the cuticle which increase $R$ will decrease $L$, stiffness and bulk modulus. These features are in qualitative agreement with experimental observations (*Park et al., 2007*; *Gilpin et al., 2015*).

Quantitatively, we observe that : (i) the length $L$ of the mutants in *Park et al. (2007)* is systematically larger than our predictions; (ii) the radius of *lon-2* mutants in *Park et al. (2007)*, which is about 25% smaller than the wild type, cannot be obtained in our simulations, as seen in *Appendix 11—figure 1C*. (i) may be due to the likely non-isotropy of the Young's modulus generated by the annular structure of the external layers. That is expected to make the stiffness smaller in the longitudinal than the orthogonal direction. (ii) is due to the fact that $R$ cannot be smaller than the value for the unpressurized shell, which is only 20% smaller than the wild type. Since relative changes in $R$ grow with $p/E$, it is likely that uncertainties on $R$ are due to fluctuations in $p/E$ among worms (see main text). As already mentioned above, both $R$ and $L$ could also be affected by our neglecting developmental effects.

We finally compare in *Appendix 11—figure 1C-D* our predictions to experimental data from *Park et al. (2007)* and *Gilpin et al. (2015)*. The agreement is notable and leads to the prediction that the bulk modulus in *lon-2* mutants should significantly deviate from the wild type. This differs from the conclusion that the bulk modulus is not affected by the cuticle (*Gilpin et al., 2015*), which was based on mutants other than *lon*-2.

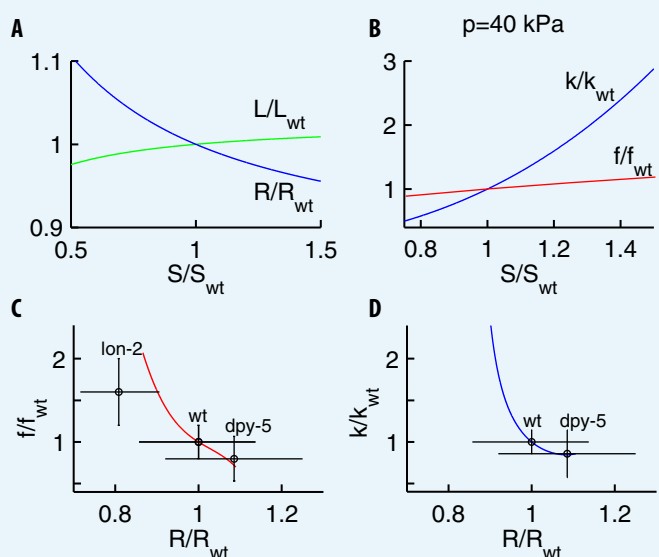

**Appendix 11—figure 1.** Changes in the mechanics caused by mutations of proteins in the cuticle. Mutations are modeled via changes of the stretching stiffness $S$ with respect to the wild type $S_{wt}$. (**A**) As $S$ increases, the geometry of the pressurized cylinder modifies: its length $L$ increases and its radius $R$ decreases. (**B**) As $S$ increases, the bulk modulus $k$ (blue) and the stiffness $f$ (red) of the force-indentation relation increase. (**C–D**) Our theoretical predictions and experimental measurements (*Park et al., 2007*; *Gilpin et al., 2015*) for $f$ and $k$ vs $R$. Since the radius of the mutants was not reported in *Gilpin et al. (2015)*, we used values in *Park et al. (2007)*. The upshot is that *lon*-2 mutants should have a bulk modulus significantly different than the wild type.

DOI: https://doi.org/10.7554/eLife.43226.036

## Appendix 12

DOI: https://doi.org/10.7554/eLife.43226.016

### Inference of the model parameters

The amplitude of $\Gamma(0)$, which controls the scale of the elongation in *Equation (5)*, is set to unity by redefining the parameters $g_h$ and $g_s$ in *Equation (8)*. As for the rates $R$, a parsimonious form that respects *Equations (7) and (10)* is

$$R_{sc} = r_{cs}e^{(1+b)(1-a)\beta\Delta G_{oc}}; \ R_{cs} = r_{cs}e^{b(1-a)\beta\Delta G_{oc}}; \tag{46}$$

$$R_{os} = r_{so}e^{(1+d)\beta\Delta G_{oc}}; \ R_{so} = r_{so}e^{da\beta\Delta G_{oc}}; \tag{47}$$

Here, $r_{cs}$ ($r_{so}$) controls the rate of the transitions between the closed and the subconductance states (the subconductance and the open states) and the parameters $b, d \in [-1, 0]$ control their global shift with respect to variations of the free-energy difference. Parameters are inferred from experimental curves by using the MATLAB optimization function 'lsqcurvefit', based on the least-squares distance between the predicted and the observed current profiles. For every realization of the quenched disorders (distribution of the channels and the initial direction of their filaments), we obtain the best parameters, which are then averaged. Their means are $\tau = 1.4\,\text{ms}$; $g_h = 1.4\,10^{-3}$; $gs/gh = 0.09$; $r_{cs} = 1/69.5\,\text{ms}$; $b = -0.75$ ; $r_{so} = 1/18\,\text{ms}$; $d = -0.56$; $a = 0.50$; $i_s/i_0 = 0.71$.

## Appendix 13

DOI: https://doi.org/10.7554/eLife.43226.016

### Dependence of neural responses on the properties of elastic filaments

This Appendix will investigate the dependence of the TRN neural response on the interaction between the channels and the surrounding medium, which we have embodied here into an elastic filament. In our model, those interactions are described by two parameters: the elastic constant and its friction coefficient with the medium. The neural response depends on the ratio $\tau = \gamma/k$, which appears in *Equation (5)* of the main text. To understand the effects of $\tau$, we computed the TRN current predicted by our model in response to a step of fixed amplitude as a function of $\tau$, keeping fixed all other parameters detailed in the main text. *Appendix 13—figure 1* shows the peak current and the decay time, i.e. the time for the current to reach half of its peak value, both averaged over the statistical realizations of the distributions for the channels. Both the peak current and the decay time increase with $\tau$: filaments relax more slowly, which provides a stimulus on the associated channel that lasts longer and thereby allows for a higher current. detailed in the main text. *Appendix 13—figure 2* shows the histogram of least-squares errors for individual responses to the stimuli in *Figure 3* of the main text for different geometries of the filaments attached to the channels. The black curve shows that individual realizations as well (and not just the average as in the main text) reproduce neural responses for the various profiles, strengths and frequencies. The figure also presents results for a model with filaments initially or permanently restricted to be tangential. Graphs indicate that our predictions are further improved by introducing those additional assumptions.

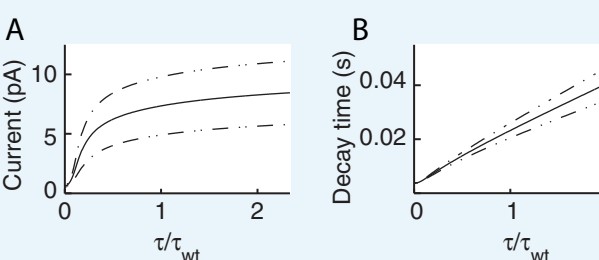

**Appendix 13—figure 1.** Dependence of the neural response on the filament-medium interaction. Peak current (**A**) and decay time (**B**) as a function of the relaxation time $\tau$ of the elastic filament connected to the channel. The model predicts that both the peak current and the decay time increase with $\tau$.

DOI: https://doi.org/10.7554/eLife.43226.039

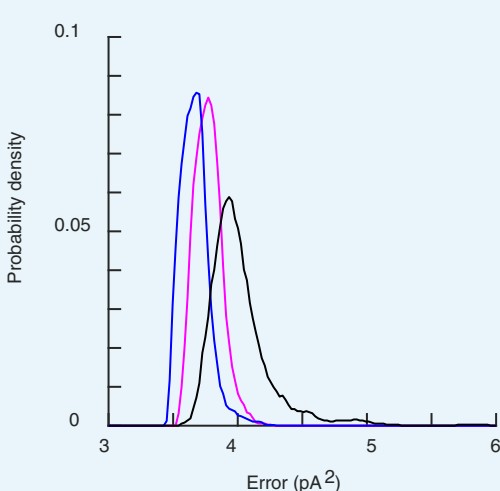

**Appendix 13—figure 2.** Dependence of the neural response on the geometry of the filaments. The error for the profiles of stimulation in *Figure 3* of the main text. The black histogram is built from individual realizations for the unconstrained model discussed in the main text. The purple and the blue curves refer to the corresponding histograms for filaments initially or permanently restricted to be tangential to the neural membrane.
DOI: https://doi.org/10.7554/eLife.43226.040

