## [Decision Letter]

[Editors’ note: this article was originally rejected after discussions between the reviewers, but the authors were invited to resubmit after an appeal against the decision.]

Thank you for submitting your work entitled "Tissue mechanics and somatosensory neural responses govern touch sensation in *C. elegans*" for consideration by *eLife*. Your article has been reviewed by two peer reviewers, and the evaluation has been overseen by a Reviewing Editor and a Senior Editor. The reviewers have opted to remain anonymous.

Our decision has been reached after consultation between the reviewers. Based on these discussions and the individual reviews below, we regret to inform you that your work will not be considered further for publication in *eLife*.

The reviews below are mixed, with one reviewer strongly in favour of the work and the second raising issues of the ultimate significance of the work, in light of a previous publication (Eastwood et al.). In our consultations we concluded that the comments of this reviewer do indeed argue against publication in *eLife*. Having said this, the work was viewed as high quality and deserving of publication in a more specialized journal where this kind of follow-up study would be appropriate.

*Reviewer #1:*

This impressively thorough and scholarly manuscript extends, buttresses, and refines this collaboration's earlier studies on the biophysics of mechanosensation in *C. elegans*. Their earlier study developed a microscopic understanding of force-displacement-current relationships of the core mechanosensitive element in *C. elegans* touch receptor neurons. Here, they put these elements in the context of a minimal model of the worm body, its cylindrical shape, the elasticity of cuticle, and its internal pressurization.

The mechanical model is elegant in its simplicity. As happens in the theory of elasticity, it is difficult to break down in analytical solutions. However computational techniques and intuitive reasoning are able to explain a surprisingly large range of phenomenology in the mechanical responsiveness of TRNs across different experiments and parameters.

It is difficult for me to "check the math" of all of their claims, but the logic of their arguments is uniformly persuasive and clear. Any reader will come away with a deeper appreciation of the biophysics of mechanosensation, which this group is systematically elevating, in its rigor and insightfulness, to the remarkably heights achieved by the hair cell literature.

In summary, the thoroughness and care of the entire manuscript is truly remarkable. I have no comments to further improve what I think is a first-rate study.

*Reviewer #2:*

In this manuscript the authors describe a quantitative model of gentle touch mechanotransduction in *C. elegans*, incorporating the mechanical deformation of the worm's body and the subsequent gating of currents in touch receptor neurons. The work extends the model previously proposed by some of the same authors in Eastwood et al. (2015) by providing more detail about how the mechanotransduction process may occur. The proposed model is novel, mostly plausible, and generally agrees with the experimental data.

I have some concerns about the assumptions made in the model (see detailed comments below).

However my main concern is that while the model fills in some quantitative details, it does not offer any significant new insights into the system. The main result summarized in the Abstract is "Our model demonstrates how the onset-offset symmetry arises from the coupling of mechanics and adaptation, and accounts for experimental neural responses to a broad variety of stimuli." But the onset-offset symmetry was previously explained in Eastwood et al.; the present paper just fleshes out the model a bit more. Similarly, the "accounting" for various experimental stimuli provides very little new understanding that was not already there. There are no resolutions of important outstanding questions, or surprising predictions, or glimpses of new phenomena.

In my opinion the manuscript would be better suited for a more specialized journal, perhaps one devoted to biomechanics and/or computational modeling.

– The title of the manuscript is rather non-specific and differs only slightly from that of a previous paper by some of the authors, reinforcing the impression that the ideas presented here are not really new.

– Subsection *“C. elegans* body mechanics”: *"*the latter [tube] is formed by pharynx, intestine, and gonad" is very inaccurate. While the pharynx and intestine form a single tube, the gonad is completely distinct from the digestive tract and moreover does not resemble a single tube, being doubled over on both ends.

– Subsection *“C. elegans* body mechanics”: What exactly is meant by an "effective description"? Would it be adequate to simply say "description"?

– Subsection *“C. elegans* body mechanics”: "Natural stimuli substantially indent the body of the worm; typical experiments feature a bead of radius 5-10 µm…" Bead indentations are not natural stimuli!

– Subsection *“C. elegans* body mechanics”: "strain within the shell is modest yet the displacement of material points can be substantial". Words like "modest" and "substantial" are vague and need to be clarified quantitatively.

– Subsection “The mechanical elastic model”, fifth paragraph: The hypothesis of constant internal pressure during deformation is poorly supported. Can the authors estimate how much change in pressure would occur given the deformations and worm mechanical properties? This would provide a firmer basis for this assumption.

– Subsection “The opening/closing dynamics of channels”: "…to reduce free parameters, its free energy is assumed intermediate between the closed and open state…" Reduction of free parameters is a not a sound basis for assuming a specific value for a parameter. The authors should provide some basis for this assumption.

– Subsection “Experimental validation of the mechanical model”, first paragraph: One can write that the mutation alters the cuticle, or the cuticle is altered in this mutant, but it's not correct that the mutant alters the cuticle.

– Subsection “Experimental validation of the mechanical model”, fifth paragraph: 40kPa is not really "consistent with" a range 2-30 kPa. I suggest "… is comparable to…"

– Subsection “Experimental validation of the mechanical model”, eighth paragraph: "Predictions are valid for all profiles of stimulations". It's unclear what "valid" means here. I suggest "Predictions are possible for all profiles of stimulation".

– Subsection “Experimental validation of the mechanical model”, eleventh paragraph: "response to sinusoidal stimuli roughly oscillates at twice the input frequency…" Why would it not be exactly twice the input frequency?

– Subsection “Experimental validation of the mechanical model”, eleventh paragraph: "Quantitative aspects are also well captured by our predictions". What exactly does the capture of aspects by predictions mean? The authors should be as specific as possible.

– Subsection “Explaining the effect of dissection protocols on neural responses”, second paragraph: Why does the internal pressure not go to zero when the worm is punctured?

– Subsection “Shell bending is weak compared to stretching; stiffness is dominated by internal pressure”: "The functions… are empirically extracted from simulations as:" The authors need to show much more detail here. If this is an empirical fit to the simulations, we need to see the results to assess the goodness of fit.

– Subsection “The onset/offset symmetry”: "components… are small" should be made more specific. Small compared to what?

– Subsection “Neural response at the single channel level: isotropic vs. directional models”, last two paragraphs: This section on potential future experiments would be much stronger if the authors could make specific, testable predictions for each experiment based on the model.

– Figure 9C: How is mean error defined and why is it very close to 1 over nearly its entire range? What are the units of measurement?

– In the Discussion, the authors state that "Our model explains several aspects of the coupling between mechanics and neural responses" and proceed to list 5 points (comments below).

1) "First, we capture the experimental current… in response to… indentation profiles…" If "capture" means "model", this sentence is just a recapitulation of the goals of the work. If it means something else, the authors should clarify.

2) "…the model explains the.… observation that neural responses are similar for displacement-clamped stimulations…" As explained in Eastwood et al., this finding is a direct consequence of a mechanosensitive channel opening dependent on strain. Therefore this result is an assumption in this model, not a result of the model.

3) "shows that internal pressure… is the major source of variability among different animals". The evidence provided for this is suggestive at best. Why should there be any internal pressure difference at all in a punctured animal?

4) "the model predicts how the neural response should change with the size of the indenter" and;

5) "…the variation of deformation evidences… a dependence on the angular position of the TRN…": These are predictions, not explanations.

– Discussion section: "…bending, an ingredient necessary for swimming". Considering that all locomotion in *C. elegans* occurs by bending the body, it is odd that swimming in particular is singled out here.

---

## [Author Response]

[Editors’ note: the author responses to the first round of peer review follow.]

Reviewer #2:In this manuscript the authors describe a quantitative model of gentle touch mechanotransduction in C. elegans, incorporating the mechanical deformation of the worm's body and the subsequent gating of currents in touch receptor neurons. The work extends the model previously proposed by some of the same authors in Eastwood et al. (2015) by providing more detail about how the mechanotransduction process may occur. The proposed model is novel, mostly plausible, and generally agrees with the experimental data.I have some concerns about the assumptions made in the model (see detailed comments below).However my main concern is that while the model fills in some quantitative details, it does not offer any significant new insights into the system. The main result summarized in the Abstract is "Our model demonstrates how the onset-offset symmetry arises from the coupling of mechanics and adaptation, and accounts for experimental neural responses to a broad variety of stimuli." But the onset-offset symmetry was previously explained in Eastwood et al.; the present paper just fleshes out the model a bit more. Similarly, the "accounting" for various experimental stimuli provides very little new understanding that was not already there. There are no resolutions of important outstanding questions, or surprising predictions, or glimpses of new phenomena.

Eastwood et al. introduced a model that accounted for the onset/offset symmetry of touch response but made the major simplification of replacing the ensemble of channels along the Touch Receptor Neurons (TRNs) by a single effective channel. This simplification neglected the spatially-varying nature of the deformation profiles along the body of the worm and the recruitment of channels known to be distributed along the neuron. Similarly to a mean-field approximation in physics, the simplification provided useful insight but had crucial limitations, both theoretical and predictive. Theoretical limitations were major, as the nonlinear mechanics of the body of the worm was not accounted. For predictive purposes, replacing the number of recruited channels by an effective factor in the strength of a single channel works well for relatively simple profiles of stimulation but it is substantially limiting for predicting responses to realistic stimuli. In particular, stimuli with large or divided contact areas, or stimuli with varying timescales, would not be well described.

The current manuscript fills the above major gaps and illustrates previous limitations by discussing pre-indentation experiments that were not previously reported. New data reveal that the combination of skin and neurons responds to pre-indentation with increased currents rather than fully adapting to persistent stimulation. The effect is entirely due to the spatial effects dealt with in this paper, and could not be captured by previous modeling or by classical models of other mechanosensory cells such as vertebrate hair cells, as now made explicit in Figure 4. These experiments provide a simple, yet striking illustration of the general point brought by this paper that touch sensation cannot be fully understood without properly accounting for the dynamics of skin deformations that lead to the activation of multiple ion channels along the neurons. Furthermore, the nonlinear mechanics of the body of the worm developed here allowed us to resolve issues in the literature on disparately inconsistent values of elastic parameters, elastic regimes of worm’s body, and the role of mutations in proteins composing the cuticle.

We have replaced title, Abstract and substantially rewritten the Introduction and the Discussion. We have taken into account all the other comments by reviewer 2.The title of the manuscript is rather non-specific and differs only slightly from that of a previous paper by some of the authors, reinforcing the impression that the ideas presented here are not really new.

Title and Abstract have been replaced.

– Subsection “C. elegans body mechanics”: "the latter [tube] is formed by pharynx, intestine, and gonad" is very inaccurate. While the pharynx and intestine form a single tube, the gonad is completely distinct from the digestive tract and moreover does not resemble a single tube, being doubled over on both ends.– Subsection “C. elegans body mechanics”: What exactly is meant by an "effective description"? Would it be adequate to simply say "description"?– Subsection “C. elegans body mechanics”: "Natural stimuli substantially indent the body of the worm; typical experiments feature a bead of radius 5-10 µm…" Bead indentations are not natural stimuli!

The wording has been modified.

– Subsection “C. elegans body mechanics”: "strain within the shell is modest yet the displacement of material points can be substantial". Words like "modest" and "substantial" are vague and need to be clarified quantitatively.

We have included a reference to Appendix A, where quantification was provided.

– Subsection “The mechanical elastic model”, fifth paragraph: The hypothesis of constant internal pressure during deformation is poorly supported. Can the authors estimate how much change in pressure would occur given the deformations and worm mechanical properties? This would provide a firmer basis for this assumption.

As discussed below, the internal pressure arises from the dynamics and effects of internal organs contained within the stiff cuticle. Modeling these effects requires an additional knowledge of the contribution of each tissue. Such a model would be quite independent of the current model and would be a significant undertaking in and of itself. We take this assumption as a working hypothesis and its justification is empirical. The text has been made more explicit on this point.

– Subsection “The opening/closing dynamics of channels”: "…to reduce free parameters, its free energy is assumed intermediate between the closed and open state…" Reduction of free parameters is a not a sound basis for assuming a specific value for a parameter. The authors should provide some basis for this assumption.

In the absence of more specific information, Occam’s razor seems a sound and well-established way of proceeding. Its empirical consistency is provided by comparison with experimental results. We have reformulated the statement and justification in the text.

– Subsection “Experimental validation of the mechanical model”, first paragraph: One can write that the mutation alters the cuticle, or the cuticle is altered in this mutant, but it's not correct that the mutant alters the cuticle.

The reviewer is correct. Thank you for bringing this to our attention, we apologize for the elision between genotype and phenotype. We have revised the sentence.

– Subsection “Experimental validation of the mechanical model”, fifth paragraph: 40kPa is not really "consistent with" a range 2-30 kPa. I suggest "… is comparable to…"

Revised to read “The value of 40 kPa is on the same order of magnitude as the range of pressures measured in the larger nematode *Ascaris lumbricoides* [Harris and Crofton, 1957].”

– Subsection “Experimental validation of the mechanical model”, eighth paragraph: "Predictions are valid for all profiles of stimulations". It's unclear what "valid" means here. I suggest "Predictions are possible for all profiles of stimulation".

Simulations developed from this model recapitulate responses that could not be addressed by our prior model. This is now made explicit.

– Subsection “Experimental validation of the mechanical model”, eleventh paragraph: "response to sinusoidal stimuli roughly oscillates at twice the input frequency…" Why would it not be exactly twice the input frequency?

Because the original frequency is still excited and present in the response, and because nonlinearities and delays produce also higher frequencies. We have clarified this point in the text.

– Subsection “Experimental validation of the mechanical model”, eleventh paragraph: "Quantitative aspects are also well captured by our predictions". What exactly does the capture of aspects by predictions mean? The authors should be as specific as possible.

Revised to “Simulations also capture the empirical relationship between speed and current amplitude (Figure 3C).”

– Subsection “Explaining the effect of dissection protocols on neural responses”, second paragraph: Why does the internal pressure not go to zero when the worm is punctured?

Contrary to molecules in a gas-filled balloon, the tubular internal organs (intestines and gonad) of a worm are much larger than the opening created by a puncturing. Some portion of the organs are released outside the animal, in a manner similar to the intestinal inversion seen in *him-4* mutants, but the majority of them cannot pass through the small incision. The majority of organs are then pushed up against the opening, and largely seals the hole. The internal pressure defined in the model is a combination of hydrostatic pressure retained by the partial seal and the resistance to compression due to the remaining organs. Although individual dissections vary in the fraction of the organs that escape the body cavity (which is consistent with the large variability discussed in our paper), we observe that, following the initial dissection, there is still more than enough pressure to push neuronal cell bodies out through a second, very small incision made in the cuticle. A detailed discussion of these points is now included in the text.

– Subsection “Shell bending is weak compared to stretching; stiffness is dominated by internal pressure”: "The functions… are empirically extracted from simulations as:" The authors need to show much more detail here. If this is an empirical fit to the simulations, we need to see the results to assess the goodness of fit.

We have included the following information:

“The above functions 𝒢_3_ and 𝒢_1_ are determined as follows. We computed numerically the force indentation relation of cylinders of different radius (R=25, 40, and 50μm) to stimulations produced by beads of different size (*R_b_* from 3 to 10μm); results of the simulations are then used to fit coefficients of the Taylor expansions of 𝒢_1_(*x*) and 𝒢_3_(*x*). Using this approach we find that the functional form

F=α1pRω01+α2RbR1+α3ω0R, (12)

with α_1_ = 0.76$, α_2_ = 2.1$, and α_3_ = 0.66, captures quantitatively the behavior of the force indentation relation (Figure 7B, R^2^= 0.995).”

We could also include the figures of the fits if explicitly requested.

– Subsection “The onset/offset symmetry”: "components… are small" should be made more specific. Small compared to what?

Small compared to the rest of the components of the tensor, namely the tangential ones. This is now included in the text.

– Subsection “Neural response at the single channel level: isotropic vs. directional models”, last two paragraphs: This section on potential future experiments would be much stronger if the authors could make specific, testable predictions for each experiment based on the model.

Figure 9D was meant to provide a testable prediction but the reviewer’s response made us realize that this was not quite clear so we modified the text to be more specific and explicit.

– Figure 9C: How is mean error defined and why is it very close to 1 over nearly its entire range? What are the units of measurement?

There was an error in the y-label, which has now been corrected. Thanks for pointing this out.

– In the Discussion, the authors state that "Our model explains several aspects of the coupling between mechanics and neural responses" and proceed to list 5 points (comments below).

Thanks for these detailed critiques.

1) "First, we capture the experimental current… in response to… indentation profiles…" If "capture" means "model", this sentence is just a recapitulation of the goals of the work. If it means something else, the authors should clarify.2) "…the model explains the.… observation that neural responses are similar for displacement-clamped stimulations…" As explained in Eastwood et al., this finding is a direct consequence of a mechanosensitive channel opening dependent on strain. Therefore this result is an assumption in this model, not a result of the model.

We had previously observed variations in response to steps of fixed indentation or force across worms, with responses being more similar if plotted against indentation than force. Here, we showed that this empirical observation can be explained in terms of the effects of internal pressure on local strain. This result requires a quantitative understanding of how body mechanics affects stimuli on the channels and, as shown in Figure 5B, the spatial component of the body deformation plays a key role and the effect would not be captured by the mean-field approach of Eastwood et al., where the body mechanics and effects of deformation profile are not included in the model.

3) "shows that internal pressure… is the major source of variability among different animals". The evidence provided for this is suggestive at best. Why should there be any internal pressure difference at all in a punctured animal?

See comments above for more detail. In brief, a pressure difference remains following the dissection procedure because only a subset of internal organs are everted during dissection through the small incisions we make in the cuticle.

4) "the model predicts how the neural response should change with the size of the indenter" and;5) "…the variation of deformation evidences… a dependence on the angular position of the TRN…" These are predictions, not explanations.– Discussion section: "…bending, an ingredient necessary for swimming". Considering that all locomotion in C. elegans occurs by bending the body, it is odd that swimming in particular is singled out here.

The Discussion section has been modified to accommodate the reviewer’s remarks and to better highlight the novelty of our results.